# An mRNA-binding channel in the ES6S region of the translation 48S-PIC promotes RNA unwinding and scanning

**Irene Díaz-López[1], René Toribio[2†], Juan José Berlanga[1], Iván Ventoso[1]\***

[1]Centro de Biología Molecular "Severo Ochoa" (CSIC-UAM), Departamento de Biología Molecular, Universidad Autónoma de Madrid (UAM), Madrid, Spain; [2]Centro de Biotecnología y Genómica de Plantas, Madrid, Spain

**Abstract** Loading of mRNA onto the ribosomal 43S pre-initiation complex (PIC) and its subsequent scanning require the removal of the secondary structure of the by RNA helicases such as eIF4A. However, the topology and mechanics of the scanning complex bound to mRNA (48S-PIC) and the influence of its solvent-side composition on the scanning process are poorly known. Here, we found that the ES6S region of the 48S-PIC constitutes an extended binding channel for eIF4A-mediated unwinding of mRNA and scanning. Blocking ES6S inhibited the cap-dependent translation of mRNAs that have structured 5′ UTRs (including G-quadruplexes), many of which are involved in signal transduction and growth, but it did not affect IRES-driven translation. Genome-wide analysis of mRNA translation revealed a great diversity in ES6S-mediated scanning dependency. Our data suggest that mRNA threading into the ES6S region makes scanning by 48S PIC slower but more processive. Hence, we propose a topological and functional model of the scanning 48S-PIC.

**\*For correspondence:**
iventoso@cbm.csic.es

**Present address:** [†]Centro de Biotecnología y Genómica de Plantas, Madrid, Spain

**Competing interests:** The authors declare that no competing interests exist.

## Introduction

Translation initiation of eukaryotic mRNA generally follows the scanning mechanism, which starts with the attachment of an activated 40S ribosomal subunit (43S preinitiation complex; PIC) to the cap structure of mRNA (forming the 48S-PIC), and the movement of the resulting complex along the 5′ UTR of mRNA to locate the initiation codon (TIS, generally AUG). This process directly depends on the extent to which a particular mRNA recruits the PIC, but also on the degree of RNA secondary structure (classical dsRNA and G-quadruplexes) found in its 5′ UTR (*Hinnebusch, 2011*; *Jackson et al., 2010*; *Kozak, 1989*; *Murat et al., 2018*; *Parsyan et al., 2011*; *Pelletier and Sonenberg, 1985*). To locate the TIS, the 48S-PIC must inspect the mRNA sequence codon-by-codon in the decoding groove of 40S neck (also known as the mRNA channel), which includes P and A sites for aminoacyl-tRNA binding. Thus, mRNA molecules must enter the channel in a single-stranded form, which requires prior unwinding by the RNA helicases associated with the 48S-PIC.

eIF4A is the main helicase that removes local secondary structure by alternating cycles of binding and dissociation from the mRNA during the scanning process (*Parsyan et al., 2011*; *Rogers et al., 1999*; *Svitkin et al., 2001*). eIF4A binds eIF4G to form the eIF4F complex together with eIF4E, which collectively promotes activation of the mRNA, recruitment of the PIC and the scanning process (*García-García et al., 2015*; *Nielsen et al., 2011*; *Pestova and Kolupaeva, 2002*; *Rogers et al., 1999*). Recently, other RNA helicases such as yeast Ded-1 (and perhaps its mammalian ortholog DDX-3), DHX-29, DHX-36 and DHX-9 have been reported to assist in the RNA unwinding and scanning of specific mRNA subsets (*Gao et al., 2016*; *Guenther et al., 2018*; *Gupta et al., 2018*; *Murat et al., 2018*; *Pisareva et al., 2008*). It is thought that eIF4F binding to the cap removes the eventual secondary structure near the 5′ extreme of the mRNA (activation) to favor recruitment

of the PIC. Recent data support the notion that mRNA is threaded into the channel from the solvent side of the 40S subunit through a chain of cooperative interactions involving eIF4E-eIF4G-eIF3-40S (*Kumar et al., 2016*). The scaffold protein eIF4G can bind both eIF3 (as part of the PIC) and the 40S subunit itself near the feet of the solvent side, thus promoting the attachment of PIC to mRNA (*LeFebvre et al., 2006*; *Villa et al., 2013*; *Yu et al., 2011*). However, little is known about the topology of the 48S-PIC, in part because its presumably dynamic nature has limited the use of cryo-EM reconstruction, which has successfully resolved more stable complexes including the PIC (*Erzberger et al., 2014*; *Hashem et al., 2013*; *Marintchev et al., 2009*). Recently, the helicase eIF4A has been located bound to alphaviral mRNA on the solvent side of the 48S-PIC assembled in vitro (*Toribio et al., 2018*). This would support a scanning complex model in which eIF4A is placed at the leading edge of the 40S subunit, thus 'pulling' the complex forward as it advances along the mRNA (*Marintchev et al., 2009*; *Toribio et al., 2018*). This model would also explain the greater dependence on eIF4A activity of mRNAs that have structured or long 5′ UTRs, many of which are involved in cell-cycle regulation and proliferation (*Modelska et al., 2015*; *Rubio et al., 2014*; *Wolfe et al., 2014*). For this reason, natural inhibitors of eIF4A are currently being tested as anti-cancer drugs (*Chu and Pelletier, 2015*).

The existence of RNA extensions such as ES6S and ES3S protruding from the solvent side of the 40S body, specifically in mammals, has been interpreted as a platform to recruit eIFs and other ligands, as described recently for ES27L of the 60S subunit (*Fujii et al., 2018*; *Knorr et al., 2019*), although direct support for this idea has not yet been presented. The ES6S region is composed of four RNA helices near the feet of the 40S particle, which form a bundle of tentacle-like structures (*Anger et al., 2013*; *Lomakin and Steitz, 2013*; *Melnikov et al., 2012*). Two of these helices (ES6S[A] and ES6S[B]) project outward from the ribosome body, and comparative analysis of the 40S particle alone and in PIC or 80S complexes suggested that these helices can undergo some degree of conformational change (*Melnikov et al., 2012*; *Toribio et al., 2016a*). Moreover, because scanning is a unique capability of 40S that is not present in the bacterial 30S subunit, ES6S has been proposed to participate in the scanning process, at least for some viral mRNAs (*Toribio et al., 2016b*). The existence of a conserved pattern of ES6S rRNA sequence complementary with eukaryotic mRNA 5′ UTRs also suggested a role for the ES6S region in mRNA loading onto PIC (*Pánek et al., 2013*).

In previous studies, we detected the interaction of AUG-downstream nucleotides of alphaviral mRNAs with the ES6S region of the 40S ribosomal subunit (*Toribio et al., 2016a*). These viral mRNAs contain a highly stable RNA stem-loop structure (DLP) located 27–31 nt downstream of the AUG, so we were able to snapshot the eIF4A helicase bound to mRNA in 48S PICs assembled in vitro. In this work, we studied the role of ES6S in genome-wide scanning process by the 48S PIC using a combination of structural and functional analyses.

## Results

### The path of mRNA through the ES6S region of the 48S PIC

To explore the possibility that ES6S could represent a universal region of the 48S-PIC in which mRNA enters and unwinds, we systematically identified the contacts of mRNA with the 18S rRNA and 40S ribosomal proteins (RPSs) that are in or near ES6S region. To this end, we designed a synthetic 128-nt mRNA (unstructured) bearing a short (19 nt) 5′ UTR, a 3′ poly(A) tail and minimal secondary structure (*Figure 1*, see also *Supplementary file 1* for details). On the basis of previous observations with alphaviral mRNA, we introduced photo-activatable 4-thio-UTPs (4-thio-U) at positions +24, +27, +31 and +34 (downstream of the AUG) as described previously (*Figure 1a*). Sites that crosslink this mRNA with 18S rRNA were identified by reverse transcriptase termination site (RTTS) assay and RNA-seq (*Kielpinski et al., 2013*) (*Figure 1a*, left panel), which provided much higher sensitivity and coverage than the classical electrophoretic resolution in urea-polyacrylamide gels (*Toribio et al., 2016a*). We included GMP-PNP in the assays to restrict the analysis to those interactions that occurred within the 48S-PIC. The vast majority of specific crosslinking sites mapped to within the nt 700–910 region of 18S RNA, concentrating in the ES6S[E] (nt 865–910), ES6S[C–D] (nt 795–864), ES6S[B] (nt 740–793) and ES6S[A] (nt 681–735) helices (*Figure 1a*). No crosslinking with the inner residues of 18S rRNA was detected, indicating that mRNA contacted only surface-exposed regions. We found abundant contacts not only with the projected parts of ES6S, but also with

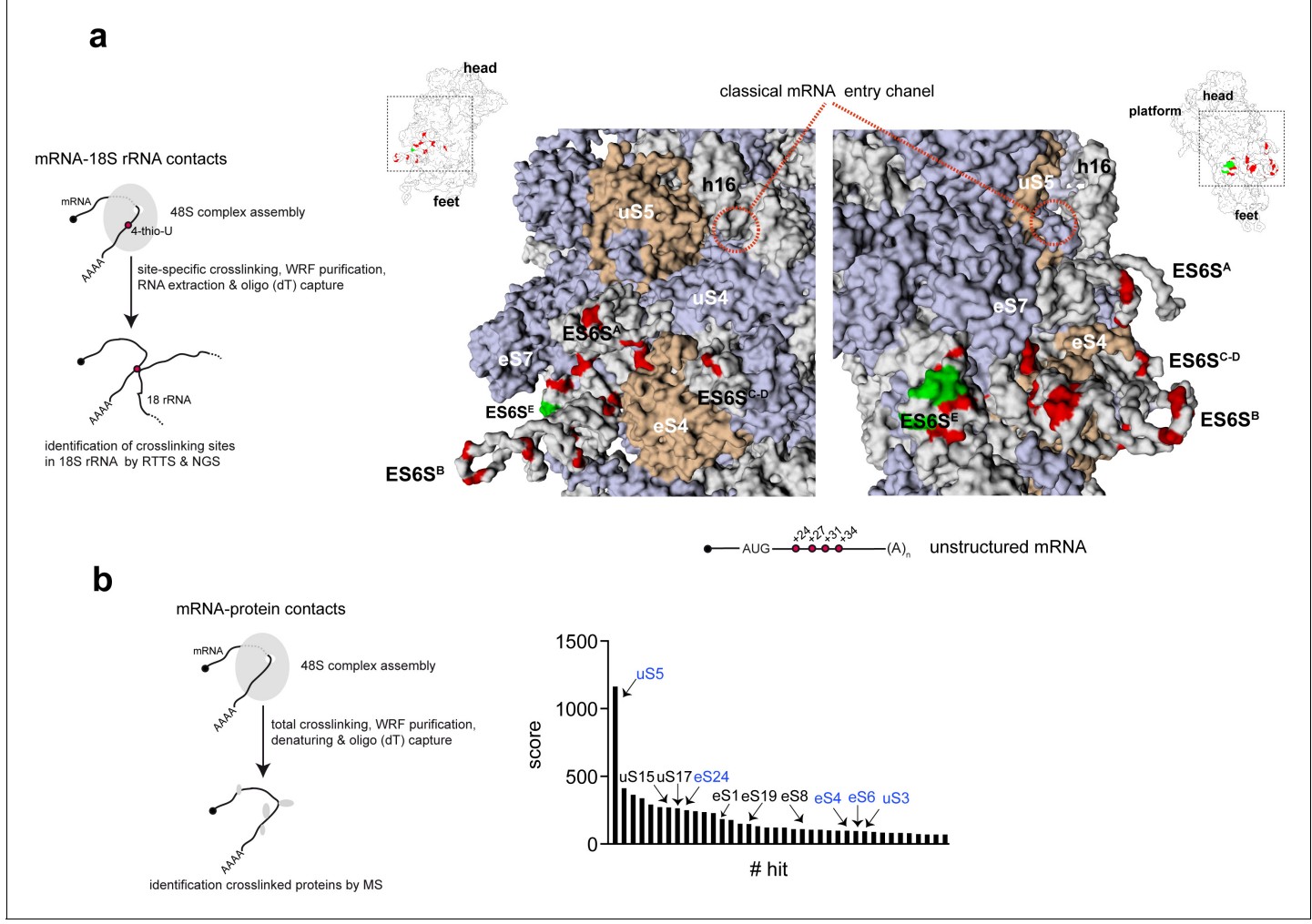

**Figure 1.** Identification of mRNA contacts with 18S rRNA and ribosomal proteins (RPSs) on the solvent side of the 48S-PIC. (a) Schematic diagram representing the method used to identify mRNA-18S rRNA contacts by specific crosslinking of 4-thio-U, followed by reverse transcriptase termination site (RTTS) assay and next-generation sequencing (NGS) (left panel). The middle and right panels show two solvent-side views of rabbit 40S (PDB: 4KZZ) that include the classical mRNA entry channel and the ES6S region. The positions of 4-thio-U residues in unstructured mRNA used for 48S PIC assembly are indicated, with the 18 most abundant crosslinking sites detected in 18S rRNA marked in red (see also *Figure 1—figure supplement 1A* for further details). Residues that have been reported to crosslink with eIF4G (*Yu et al., 2011*) are marked in green. uS5 (formerly RPS2) and eS4 (formerly RPS4X) are marked in brown, whereas the rest of the RPSs are shown in light blue. Note that, in this model, the ES6S$^A$ helix is in an 'inward' orientation towards the eIF3b. (b) Identification of RPSs that crosslink with unstructured mRNA assembled into the 48S PIC. Protein–mRNA interactions were captured throughout the mRNA (left panel); the 40 most-represented hits are shown in the right panel. Proteins marked in blue and black are those located on the solvent side and on the 60S subunit side, respectively. Note that uS5 was the hit with the highest representation.

The online version of this article includes the following figure supplement(s) for figure 1:

**Figure supplement 1.** Identification of SFV-DLP 27 mRNA contacts with 18S rRNA and RPSs on the solvent side of the 48S-PIC.

residues that are at the base of these helices (e.g. residues 802–803 of helix C), embedded in the protein–RNA layer of the 40S body (*Figure 1a*). This allowed us to draw an mRNA path through the ES6S region that extends from ES6S$^E$ to the distal loop of ES6S$^{C-D}$, including the ES6S$^B$ and ES6S$^A$ helices, and that is flanked by ribosomal proteins such as eS7, eS4 and uS4 (*Figure 1a*). In a parallel experiment, we also included a previously characterized mRNA based on the Semliki Forest virus (SFV) genome, which contains a single 4-thio-U at the 5′ flank (+27) of the DLP (*Figure 1—figure supplement 1*). SFV-DLP n27 mRNA generated a crosslinking pattern that is similar, though not identical, to that observed for unstructured mRNA. Specifically, the number of crosslinks to residues of the ES6S$^E$ and ES6S$^{C-D}$ helices increased in SFV-DLP n27 mRNA compared to unstructured mRNA (*Figure 1—figure supplement 1*).

Since the region that extends between ES6S and the classical mRNA entry channel contains no exposed 18S rRNA, a proteomic analysis of RPSs that could contact the mRNA in this region was performed. Attempts to detect proteins that are crosslinked to single 4-thio-U residues of mRNA by mass spectrometry (MS) failed, so unlabeled unstructured and SFV-DLP n27 mRNAs were used to assemble 48S-PICs that were subsequently crosslinked under a 254 nm lamp to detect RNA–protein contacts along the entire mRNA path. A similar pattern of crosslinked RPSs was observed for both mRNAs, with uS5 (formerly RPS2) the hit with the highest score and coverage (*Figure 1b* and *Figure 1—figure supplement 1*). Crosslinking of uS5 at +11 nt downstream of the AUG has been previously reported in 48S-PICs assembled in vitro using a synthetic mRNA and purified 40S subunits and eIFs (*Pisarev et al., 2008*). Interestingly, uS5 was localized between the mRNA entry channel and the ES6S region of the 40S subunit, extending about 48 Å along the channel (*Figure 1a*). We also detected other crosslinked RPSs with lower scores, including some that were located either on the solvent side of the 40S subunit (uS3, uS4, eS4, eS6 and eS24), or on the intersubunit side near the decoding site (eS19) and at the mRNA exit channel (uS17).

In previous reports, we found that both eIF4A and eIF3g could be crosslinked with alphaviral mRNAs bearing the DLP structure (*Toribio et al., 2018*). Thus, we compared the crosslinking patterns generated by unstructured mRNAs, which lacks secondary structure, and SFV-DLP mRNAs labeled with [$\alpha$-$^{32}$P] and 4-thio-U after assembly into the 48S-PIC. Owing to their similar apparent molecular weight, eIF4A and eIF3g protein bands migrated as a doublet in SDS-PAGE (*Toribio et al., 2018*). Both eIF4A and eIF3g bands were crosslinked with SFV-DLP n27 mRNA to a similar extent, whereas unstructured mRNA generated strong crosslinking with eIF3g and little (if any) with eIF4A (*Figure 2a*). This was confirmed by denaturing immunoprecipitation (dIP) experiments with specific antibodies (*Figure 2—figure supplement 1*). The fact that unstructured mRNA generated no crosslinking with eIF4A is consistent with previous data showing the critical role of DLP structure in the trapping of eIF4A within the 48S-PIC (*Toribio et al., 2018*). To map the placement of eIF4A in the 48S PIC more precisely, we systematically changed the position of 4-thio-U along the AUG-DLP stretch of SFV-DLP n27 mRNA (*Figure 2b*). Maximum crosslinking with eIF3g was observed when 4-thio-U was placed +24 nt downstream of the AUG, whereas maximum crosslinking of eIF4A was achieved when 4-thio-U was placed at +27 nt. By modeling mRNA placement throughout the ES6S region of the 48S-PIC (*Toribio et al., 2018*), we found that crosslinking of eIF3g at position +24 fit well with the suggested placement of eIF3g bound to eIF3b near ES6S$^A$ (*des Georges et al., 2015*; *Eliseev et al., 2018*; *Hashem et al., 2013*). According to our data, eIF4A may be placed a bit further downstream, probably between the ES6S$^A$ and ES6S$^B$ helices (*Figure 2c*).

## ES6S blockage inhibits translation initiation

The high copy number of rDNA genes in eukaryotic genomes makes it impossible to determine the role of the ES6S region in translation using classical genetic disruption approaches. Therefore, we decided to block ES6S by means of specific oligos targeting 18S rRNA, or by fusing RPSs surrounding the ES6S region to proteins that could sterically block this region. From among all the oligos tested, we selected oligo 4, which targets a partially single-stranded sequence in ES6S$^D$ that showed some inhibitory effects on translation of alphavirus mRNA (*Toribio et al., 2016a*). We determined the 3' effective pairings of oligo 4 with ES6S$^D$ on native 40S particles by RNAse H assay, which allowed us to construct models of oligo 4 bound to the 40S particle (*Figure 3—figure supplement 1*). In all of these models, the 5' extreme of oligo 4 is likely to be projected towards an open cavity that is delimited by ES6S$^B$, ES6S$^A$, uS19, uS4 and the C-terminus of eS4 (*Figure 3—figure supplement 1*). We also conjugated fluorophores such as VIC, Texas Red (TR) and Fluorescein (FITC) to the 5' end of oligo 4 to increase its blocking capacity and to facilitate its detection (*Figure 3—figure supplement 1*). First, we found the conditions for optimal oligo delivery into cells, which maximized transfection efficiency and reduced oligo aggregation, to increase its bioavailability (*Figure 3—figure supplement 2*). We transfected human (HEK293T, HeLa) and murine (MEF) cells with 70–90% efficiency. To confirm that oligo 4 bound to 18S rRNA properly in transfected cells, we carried out an 'in situ' RT-PCR amplification using the endogenous bound oligo 4 to prime retrotranscription of 18S rRNA isolated from transfected cells. PCR amplification of the resulting cDNA revealed that oligo 4 was bound to both 40S and 80S fractions (*Figure 3a*, lower right panel). Next, we analyzed the effect of VIC-oligo 4 on general translation by means of metabolic labeling with [$^{35}$S]-Met.

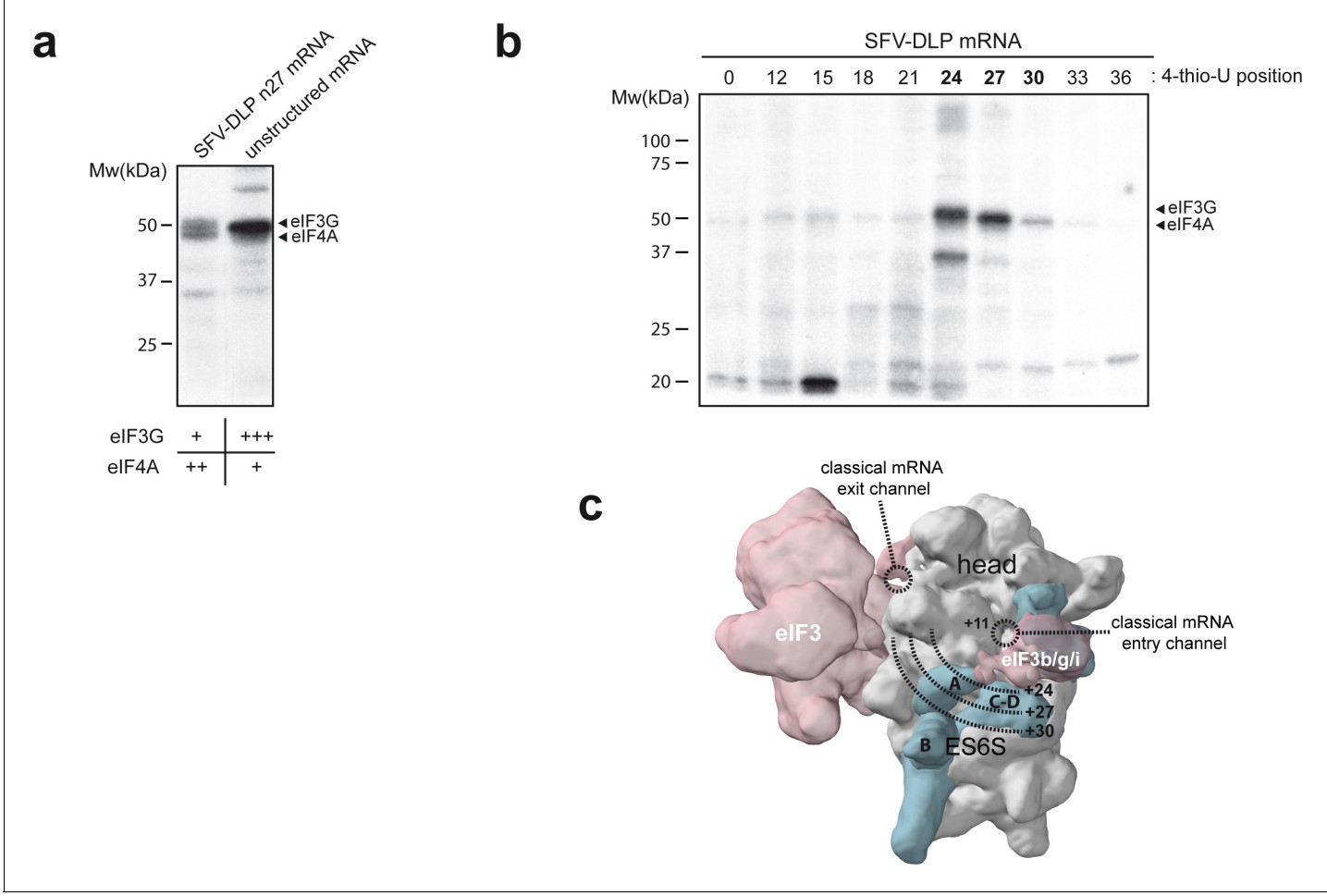

**Figure 2.** Identification of eIFs associated with mRNA in the ES6S region. (**a**) Patterns of protein crosslinking generated with the indicated mRNAs, which are labeled with 4-thio-U and [α-$^{32}$P]-GTP. The identified protein bands are indicated, with a relative quantification of the intensity of the protein bands shown below, according to this experiment and the one shown in *Figure 2—figure supplement 1*. (**b**) Mapping of the interaction of eIF4A and eIF3g with mRNA in the ES6S region. SFV-DLP mRNAs with a single 4-thio-U placed at the indicated positions were used to assemble the 48S-PIC, and the pattern of protein crosslinking was analyzed. (**c**) The positions of mRNA that crosslinked with eIF4A and eIF3g were projected on the solvent side of the PIC, assuming that the mRNA is threaded along the ES6S region. The +11 position of mRNA is placed at the mRNA entry channel (*Lomakin and Steitz, 2013*; *Pisarev et al., 2008*). A rise per base of 4.5 Å for a stretched RNA strand was used as described previously (*Toribio et al., 2016a*).

The online version of this article includes the following figure supplement(s) for figure 2:

**Figure supplement 1.** Confirmation that eIF4A and eIF3g crosslinked with mRNAs in 48S-PIC.

Transfection of MEFs with VIC-oligo 4, but not with its unconjugated form, strongly inhibited protein synthesis at 12 hr post-transfection (hpt) (*Figure 3b*, upper left panel). In these experiments, we also included oligo 5.4 (both VIC-conjugated and unconjugated), which targets helix ES6S$^B$ (*Figure 3—figure supplement 1*) without significantly affecting translation, showing that the effect of VIC-oligo 4 was specific.

To test the role of the fluorochrome in the inhibitory activity of VIC-oligo 4, we replaced VIC with FITC or TR (*Figure 3b*, upper left panel), finding a similar degree of translation inhibition for all of the fluorochromes tested. This suggests that the inclusion of an additional mass at the 5′ end was responsible for the inhibitory activity of oligo 4 on translation. The effect of VIC-oligo 4 on translation was rapid, starting at 4–6 hpt and reaching a maximum at 8 hpt (*Figure 3b*, upper right panel), suggesting a direct impact on translation rather than an indirect consequence of cell growth arrest that was induced by VIC-oligo 4 some time after post-transfection (*Figure 3—figure supplement 2*). Analysis of polysome profiles at 6–8 hpt revealed the accumulation of 80S (monosomes) together

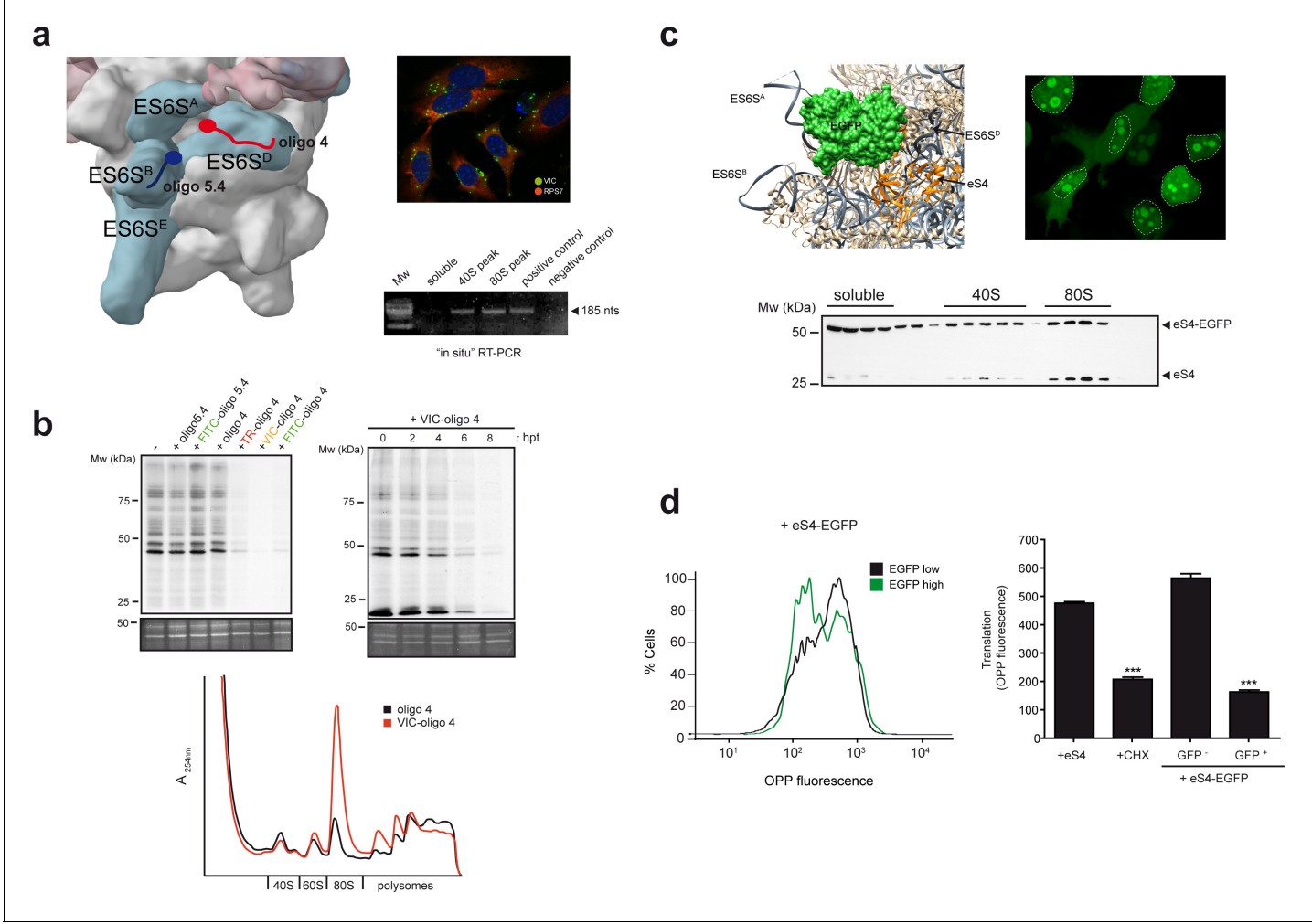

**Figure 3.** ES6S blockage affects global translation. (a) Binding site of oligonucleotides 4 (red) and 5.4 (blue) in the ES6S region of the 40S subunit (left panel). Note that in this model of PIC (EMD-5658), the ES6S[A] helix is in an 'outward' orientation. Right panels show VIC-oligo 4 uptake and binding to ES6S[D]. The upper panel shows fluorescence microscopy of VIC-oligo 4 (green) combined with an IF staining of eS7 (red) and DAPI (blue) in MEF cells. Note the typical aggregation of the oligo inside the cells. The bottom panel shows an in situ RT-PCR of RNA extracted from the indicated fractions. cDNA was primed exclusively by the VIC-oligo 4, which remained bound to ES6S[D] during RNA extraction. Total RNA from untransfected cells was included as a negative control. (b) Metabolic labeling of MEF cells with [35S]-Met. Cells transfected with the indicated oligonucleotides were labeled for 30 min at 12 hr post-transfection (hpt), and analyzed by SDS-PAGE and autoradiography. A SYPRO staining of the middle part of gels is shown as loading control (–). The upper right panel shows a time-course analysis of the VIC-oligo 4 effect on translation. The lower panel shows a polysome profile of MEFs transfected with VIC-oligo 4 (red) or with unconjugated oligo 4 (black). Extracts were separated in a 10–40% sucrose gradient and fractionated as described in the Materials and methods; the identities of the main peaks are indicated. (c) Fusion protein-mediated blockage of ES6S affects global translation. Model of EGFP (green) fused to eS4 (orange) of the 40S subunit. Note that in this model of human 40S, the ES6S[A] helix is in an 'outward' orientation (upper left panel). Fluorescence microscopy of HeLa cells expressing eS4–EGFP fusion protein (upper right panel). Micrographs were taken at 36 h. Nuclei are encircled by a dashed line; note the bright nucleolar and cytoplasmic staining. The bottom panel shows the distribution of eS4 and eS4-EGFP proteins in a 10–35% sucrose gradient from HEK293T cells transfected with eS4-EGFP. (d) Measurement of total protein synthesis by OPP fluorescence in transfected cells expressing no/low (black line) or high levels of eS4-EGFP (green line). Cells were first gated into two groups according to EGFP expression, and then the distribution of OPP fluorescence intensity was determined (left panel). The right panel shows OPP fluorescence measurements for cells transfected with eS4 alone, cells treated with 50 µg/ml CHX for 20 min, and cells transfected with eS4–EGFP that express (or do not express) EGFP. Data are the mean of three independent experiments ± standard deviations (SD).

The online version of this article includes the following figure supplement(s) for figure 3:

**Figure supplement 1.** Mapping and modeling of the oligo 4 bound to 48S-PIC.

**Figure supplement 2.** Accumulation of VIC-oligo 4 into the cells and its effect on proliferation.

**Figure supplement 3.** VIC-oligo 4 does not significantly affect 40S subunit biogenesis.

**Figure supplement 4.** Effect of VIC–oligo 4 on protein crosslinking in 48S complexes assembled with unstructured or SV-DLP n27 mRNAs.

**Figure supplement 5.** Effect of eS4–EGFP overexpression on the formation of 40S and the PIC.

with a decrease in heavy polysomes, strongly suggesting a blockade of the initiation step of protein synthesis (*Figure 3b*, lower panel). We did not detect significant alteration in the 28S/18S ratio at different times post-transfection, nor alteration in the distribution of representative RPSs across the sucrose gradient, suggesting that 40S biogenesis was not significantly altered by the oligo (*Figure 3—figure supplement 3*).

Next, we tested the effect of VIC–oligo 4 on protein crosslinking with mRNA within the 48S complex. As shown in *Figure 3—figure supplement 4*, ES6S blockage by VIC–oligo 4 drastically reduced eIF4A/eIF3g crosslinking, suggesting that VIC–oligo 4 disturbed the interaction of mRNAs with ES6S region.

We then aimed to block ES6S by fusion of heterologous polypeptides to some surrounding RPSs. Among the proteins that are located within or near the ES6S region, we selected eS4 (formerly RPS4X) as a candidate. The C-terminus of eS4 projects nearly perpendicular to the major axis of the 40S particle between ES6S$^A$ and ES6S$^D$, offering a place for the fusion of polypeptides of increasing sizes. Plasmids encoding recombinant eS4 fused to different polypeptides were transfected into HEK293T and HeLa cells, and their expression, incorporation into 40S particles and effect on translation were analyzed. Among the constructs tested, eS4 fused to EGFP was the one that gave the most consistent results. When eS4–EGFP in the 40S particle was modeled, the EGFP mass could be easily placed between ES6S$^B$ and ES6S$^{C-D}$, occupying a significant volume ($28.67 \times 10^3$ Å$^3$) (*Figure 3c*, upper right panel).

Overexpressed eS4–EGFP accumulated in the nucleolus, but also in the cytoplasm of transfected cells (*Figure 3c*, upper right panel), as previously reported for other overexpressed RPSs (*Al-Jubran et al., 2013*). Comparison of the distribution of endogenous eS4 and eS4–EGFP in sucrose gradients confirmed that eS4–EGFP was incorporated into 40S and 80S particles with the same efficiency as eS4 (*Figure 3c*, lower panel). However, we found a differential increase in the 40S accumulation of eS4–EGFP when compared to that of endogenous eS4, suggesting a block in the preinitiation step. The effect of eS4–EGFP overexpression on protein synthesis was measured by O-propargyl-puromycin (OPP) incorporation followed by fluorescent quantification of the cells expressing EGFP (or not). Clearly, protein synthesis was blocked by eS4–EGFP, to an extent comparable to that resulting from cycloheximide treatment (*Figure 3d*). To test whether the incorporation of eS4–EGFP into the 40S particle affected the binding of some eIF3 subunits (such as eIF3g, which binds eIF3b near ES6S$^A$ of 40S), we probed sucrose gradients with different antibodies (*Figure 3—figure supplement 5*), finding no significant differences in the distributions of the analyzed eIFs.

## ES6S blockage differentially affects scanning-dependent translation

ES6S may represent an mRNA-threading region for the unwinding of the RNA secondary structure that is necessary for scanning, so we evaluated the effect of VIC–oligo 4 on the translation of luciferase (luc) mRNAs bearing 5′ UTRs of different lengths (33–656 nt) and secondary structure ($\Delta G°$ from $-5$ to $-70$ kcal.mol$^{-1}$) (*Figure 4a* and *Supplementary file 1*). MEFs were cotransfected with luc plasmids and VIC–oligo 4, and luc activity was measured at 14 hpt. VIC–oligo 4 transfection reduced the accumulation of luc in all cases, but the extent of inhibition dramatically differed among the constructions tested. Translation of luc mRNAs bearing a long 5′ UTR (656 nt) or an average-sized 5′ UTR (121 nt) with a stable stem-loop structure (SL20) was dramatically impaired by VIC–oligo 4 transfection (50–80-fold inhibition) (*Figure 4a*). By contrast, translation of luc mRNA bearing a short 5′ UTR (33 nt) was much less affected by VIC–oligo 4 transfection (about two-fold inhibition). We further confirmed that sensitivity to VIC–oligo-4-induced translational block increased with the 5′ UTR length of the mRNA (*Figure 4—figure supplement 1*). We also tested the effect of VIC–oligo 4 on hepatitis C virus (HCV) IRES-driven translation, showing only a modest effect (2–3 fold inhibition), similar to that observed for mRNA with a short 5′ UTR (*Figure 4a*). To further confirm this differential effect on cap-dependent and IRES-driven translation, we tested the effect of VIC–oligo 4 on the translation of other viral mRNAs. Cells were first transfected with VIC–oligo 4, and then infected with SFV or vesicular stomatitis virus (VSV) as prototypes of cap-dependent translation, and encephalomyocarditis virus (EMC) or poliovirus (PV) as prototypes of IRES-driven translation. Accumulation of EMC or PV proteins was not significantly affected by VIC–oligo 4 transfection when compared with controls, whereas translation of SFV and VSV proteins was dramatically inhibited by VIC–oligo 4 (*Figure 4—figure supplement 1*).

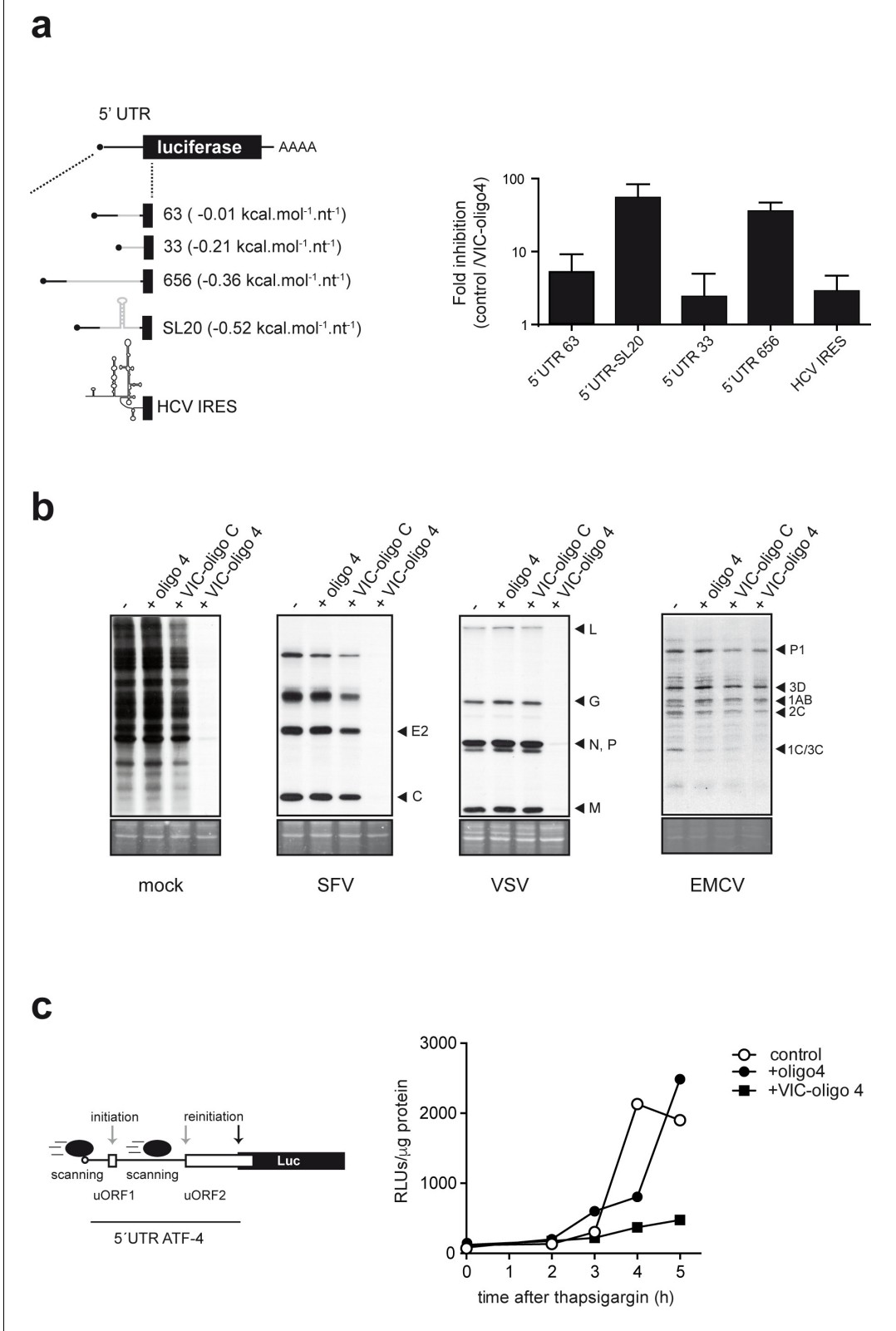

**Figure 4.** Differential effect of ES6S blockage on cap-dependent translation. (**a**) Schematic diagram of luc mRNAs bearing different 5' UTRs (details in **Supplementary file 1**). Note that the cap-proximal sequences are identical for the 63, 656 and SL20 mRNAs (denoted with a black line). HCV, hepatitis C virus. MEF cells were cotransfected with 1 μg of the indicated luc plasmid and 10 pmol of VIC–oligo 4; transfections with oligo 4 and VIC–oligo capsid did not significantly affect luc expression, and are included as controls. Data are expressed as fold inhibition with respect to oligo 4 control from

*Figure 4 continued on next page*

*Figure 4 continued*
four independent experiments (mean ± SD). (b) Effect of VIC-oligo 4 on translation of viral mRNAs. MEFs were transfected with the indicated oligonucleotides and infected 6 hr later with the indicated viruses at a multiplicity of infection (MOI) of 10 pfu/cell. At 4 hr post-infection, cells were metabolically labeled with [$^{35}$S]-Met for 30 min and analyzed as described in the Materials and methods. The positions of the main viral protein bands are indicated. (c) Effect of VIC–oligo 4 on luc translation driven by the 5′ UTR of ATF-4 mRNA. MEFs constitutively expressing 5′ UTR ATF-4–Luc mRNA were transfected with the indicated oligonucleotides, and treated 6 hr later with 2 μM thapsigargin for the indicated times.
The online version of this article includes the following figure supplement(s) for figure 4:

**Figure supplement 1.** Effect of 5′ UTR length of luc mRNAs on the sensitivity to VIC-oligo 4-mediated translational block.

As translation of ATF-4 mRNA upon stress-induced eIF2α phosphorylation is a paradigm of scanning-dependent reinitiation on downstream bona fide ATG, we tested the effect of VIC–oligo 4 on the translation of luciferase mRNA driven by the 5′ UTR of human ATF-4 mRNA. Under normal conditions, 40S initiates from the regulatory uORF1 and uORF2 in the 5′ UTR of ATF-4 mRNA, leading to constitutive translational repression that is relieved when the availability of ternary complex is reduced by eIF2α phosphorylation (*Vattem and Wek, 2004*). Clearly, VIC–oligo 4 transfection of MEFs expressing the 5′ UTR ATF-4 Luc mRNA prevented the accumulation of luc upon thapsigargin treatment, showing that VIC–oligo 4 did indeed block scanning and/or reinitiation on the 5′ UTR ATF-4 mRNA (*Figure 4c*).

## Genome-wide analysis reveals the role of the ES6S region in the translation of mRNAs with G-rich 5′ UTRs

To test the effect of VIC–oligo 4 on genome-wide mRNA translation, we carried out polysome profiling in HEK293T cells (*Figure 5a*). After transfection with oligo 4 or VIC–oligo 4, mRNA levels were quantified in monosomal (M) and polysomal (P) fractions (*Figure 5—figure supplement 1*). The P/M ratio faithfully represents the translation efficiency (TE) of a given mRNA, as we previously reported (*Ventoso et al., 2012*). For each gene, the oligo 4/VIC–oligo 4 TE ratio was determined and expressed as $\log_2$ fold change (FC TE), with those >1 considered to be downregulated ('TE down'; n = 1054) and those <−0.7 considered to be upregulated ('TE up'; n = 333). We found that the sensitivity of transcriptome-wide mRNAs to VIC–oligo 4 varied more than 30-fold, although the distribution of TE change was clearly skewed towards translation inhibition (*Figure 5a*). Very small differences in total mRNA abundance were found when comparing cells transfected with oligo 4 and VIC–oligo 4 (*Figure 5—figure supplement 1*).

Gene ontology (GO) analysis revealed that the 'TE down' group was enriched in mRNAs that are involved in cell signaling and growth, especially in the G-protein-coupled receptor signaling and Ras pathways (*Figure 5b*). This group was also enriched in KEGG pathway mRNAs related to cancer and other diseases (*Figure 5—figure supplement 1*). On the other hand, the 'TE up' group was highly enriched in GO terms related to RNA metabolism, including mRNA splicing, rRNA processing and mRNA translation (*Figure 5b*). Accordingly, the TE down group was enriched in mRNAs encoding membrane proteins and under-represented in mRNAs encoding nuclear proteins that have RNA binding or ligase activities (*Figure 5b*, lower panel).

Next, we compared some basic features of the 5′ UTR mRNA among TE groups. Principal component analysis (PCA) confirmed that 5′ UTR length, G+C composition and RNA secondary structure content were the parameters that most contributed to the variance (*Figure 5—figure supplement 2*). We found that the G+C content and the propensity to fold into secondary structures (RNAfold, ΔG°) were the 5′ UTR features that best correlated with TE groups. 5′ UTRs of the 'TE down' group showed a higher G+C content than those of the 'TE up' group (74% vs 62.96%, p=$3\times10^{-49}$, U test), and a more stable predicted RNA secondary structure (−72.50 kcal.mol$^{-1}$ vs −27.7 kcal.mol$^{-1}$, p=$3\times10^{-23}$, U test) (*Figure 5c*). The 5′ UTRs of the 'TE down' group were also larger than those of the 'TE up' group (p=$6\times10^{-12}$, U test) (*Figure 5c*). Next, we used the MEME algorithm to search for short motif enrichment in the 'TE down' and 'TE up' groups of mRNAs. Interestingly, we found a strong enrichment of 15-mer and 12-mer (GGC/A)$_4$ motifs (E-value = $2.7\times10^{-76}$ and $2.7 \times 10^{-47}$, respectively) in the 'TE down' group that was not detected in the 'TE up' group (*Figure 5—figure supplement 2*). As (GGC/A)$_4$ motifs can fold into G-quadruplexes (G4s) (*Wolfe et al., 2014*), we carried out a systematic research of the classical G4 motif in our dataset using the QuadBase2 program

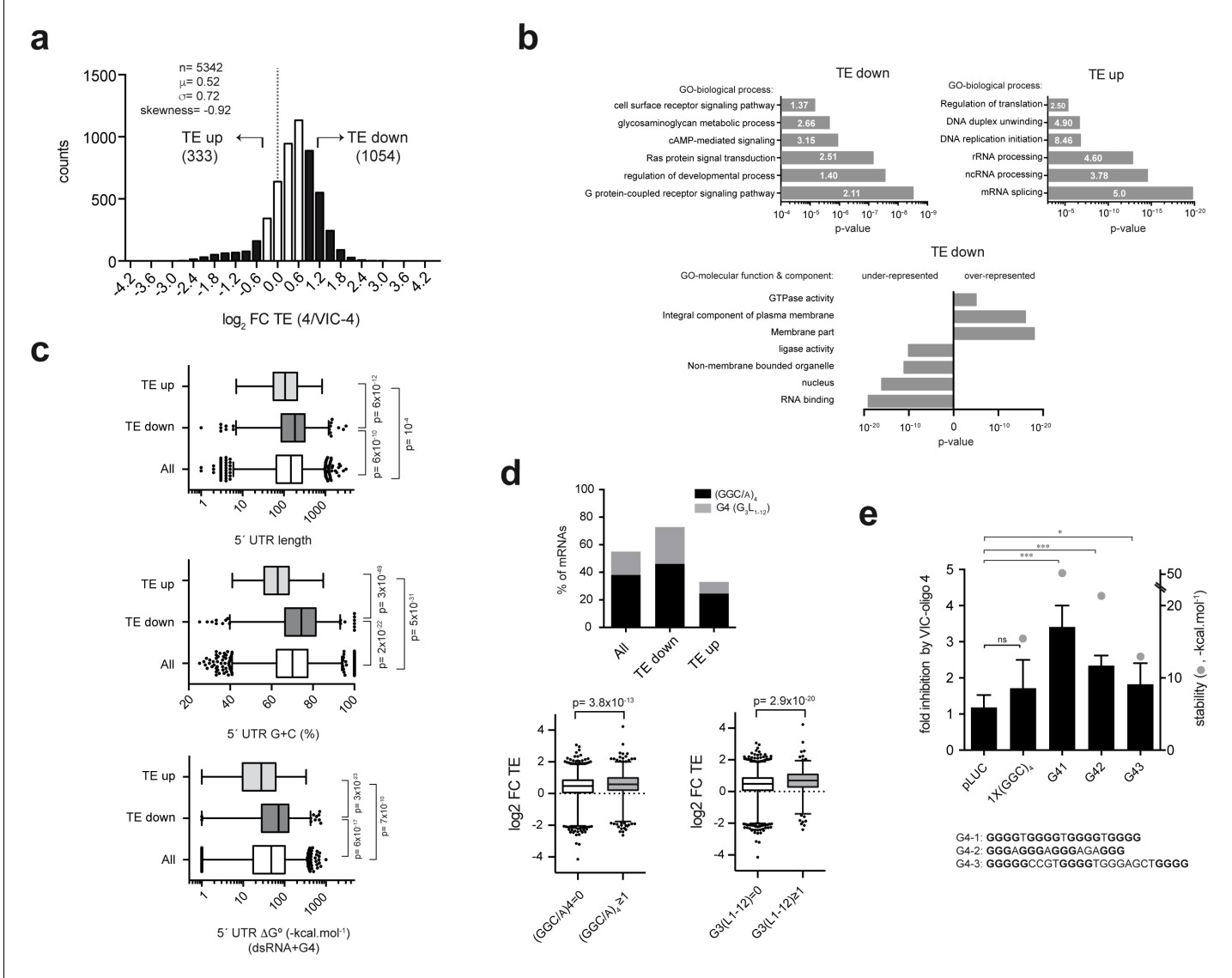

**Figure 5.** Genome-wide effect of ES6S blockage on translation. (a) Distribution of translational changes induced by VIC–oligo 4 transfection of HEK293T cells. Only mRNAs showing a p-adj <0.05 between replicas were selected; data are expressed as log2 fold change (FC) of translation efficiency (TE) (4/VIC-4). The tails of the distribution with FC TE >1 (TE down) and FC TE <−0.7 (TE up) were selected for further analysis. (b) Gene ontology (GO) analysis of the 'TE down' and 'TE up' groups, showing the six terms (biological processes) with the highest significance (p-value) and fold-enrichment (numbers within the bars) (upper panels). The lower panel shows the differential enrichment of the 'TE down' group in terms of GO-molecular function and GO-molecular component. The under- and over-represented terms are shown. (c) Analysis of the 5' UTR features of mRNAs in TE groups. The length, G+C composition and RNA structure content of 5' UTRs were analyzed and compared among groups using the Mann-Whitney U test (p-values are shown). (d) Differential enrichment in the $(GGC/A)_4$ and G4 ($G3L_{1-12}$) motifs found in the 5' UTR of 'TE down' mRNAs. Data represent the accumulated percentage of mRNAs that show at least one of the indicated motifs (left panel); p-value<$10^{-5}$ after $\chi^2$ test. The lower panel shows a box plot of the $log_2$ FC TE of mRNAs showing at least one $(GGC/A)_4$ motif (left) or one G4 ($G3L_{1-12}$) motif (right) compared with the corresponding subset of mRNA showing no motifs. The p-values generated by U tests are shown. (e) Effect of VIC–oligo 4 on the translation of Luc mRNAs bearing the indicated motifs in the 5' UTR. Translations were carried out in RRL as described in the Materials and methods. Data represent the mean ± SD of six (pLuc and 1x(GGC)₄) or three (G41, G42 and G43) independent experiments. The calculated stability of the motifs is shown as ΔG° (-kcal.mol⁻¹, gray dots); the sequences of the G4 motifs are also indicated (bottom).
The online version of this article includes the following figure supplement(s) for figure 5:

**Figure supplement 1.** Quality and reproducibility of the experiment shown in *Figure 5*.
**Figure supplement 2.** Further bioinformatic analysis of the data from experiment shown in *Figure 5*.

(*Dhapola and Chowdhury, 2016*). A strong enrichment of the $(G_3N_{1-12})_4$ motif was found in the 5′ UTRs of the 'TE down' mRNAs ($\chi^2 < 10^{-4}$), being 3-fold higher than that found in the 'TE up' group. About 75% of 'TE down' mRNAs contained either 12-mer $(GGC/A)_4$ or classical G4 (*Figure 5d*, upper left panel). To test the contribution of $(GGC)_4$ and G4 motifs to the observed translation sensitivity to VIC–oligo 4, we cloned a single copy of either motif into the 5′ UTR of a pLuc plasmid. For G4, we tested three experimentally validated variants of the motif, including one 'perfect' G4 and two motifs that are present in human Bcl2 (*Shahid et al., 2010*) and TM3-MMP (*Morris and Basu, 2009*) mRNAs (*Figure 5E*). Clearly, the presence of G4 motifs rendered the translation of luc mRNA more sensitive to VIC–oligo 4, whereas the presence of $(GGC)_4$ had less of an effect. Moreover, we found a correlation between the predicted stability of the G4 motif and the extent of translation inhibition by VIC–oligo 4 (*Figure 5e*).

Next, we selected some representative mRNAs from the 'TE down' and 'TE up' groups for validation and further analysis. Among the downregulated mRNAs, we selected CCND3, HRAS, ODC-1, AKT and GRK2, whereas eIF4B and eEF1A1 TOP mRNAs were selected as representatives from the upregulated group. The 'TE down' mRNAs had longer than average 5′ UTRs (188–395 nt) with moderate-to-strong secondary structure, including the presence of G4 or/and $(GGC)_4$ motifs (*Figure 6a*, left panel). The presence G3- and G2-quadruplexes in the 5′ UTRs of CCND3 and ODC1, respectively, has been reported before to inhibit translation (*Lightfoot et al., 2018*; *Weng et al., 2012*). By contrast, representative 'TE up' mRNAs showed shorter than average 5′ UTRs (23 and 63 nt) and lacked the secondary structure typical of 5′ TOP mRNAs (*Figure 6—figure supplement 1*). Thus, most of the 5′ TOP mRNAs detected in our dataset fell within the 'TE up' group (*Figure 6a*, right panel). Western blot analysis revealed a dramatic reduction in the accumulation of CCND3, ODC-1, HRas and AKT1 protein upon VIC–oligo 4 transfection, whereas GRK2 protein accumulation was reduced but to a lesser extent (*Figure 6b*). Time-course experiments confirmed the high sensitivity of CCND3, followed by AKT1 and HRAS, to VIC–oligo 4 (*Figure 6b*, right panel). The short half-life of the CCND3 protein (about 30 min) probably accounts for its rapid disappearance after blockage of de novo synthesis by VIC–oligo 4. However, the accumulation of eIF4B increased slightly in cells transfected with fluorophore-conjugated oligo 4s, especially TR-oligo 4 (*Figure 6c*). These results are in good agreement with the data from polysome profiling (*Figure 6—figure supplement 2*).

Translation of CCND3 has been reported to be sensitive to eIF4A inhibition (*Wolfe et al., 2014*), so we compared the effects of VIC–oligo 4 and the eIF4A inhibitor hippuristanol (hipp) (*Bordeleau et al., 2006*). Interestingly, a similar but not identical effect on translation was found (*Figure 6c*). Whereas hipp, like VIC–oligo 4, reduced the accumulation of GRK2, AKT1 and CCND3 proteins, HRAS translation was not affected by hipp, and translation of eIF4B mRNA increased upon treatment with hipp, as also observed for TR–oligo 4 (*Figure 6c*). These results suggest a partial functional overlap of the ES6S region and helicase eIF4A in the translation of specific mRNAs. To analyze this in more detail, we compared our data set with that published by *Modelska et al. (2015)* using human cells, in which the expression of eIF4A was silenced by interference. Notably, translation of these eIF4A-dependent mRNAs was also affected by VIC–oligo 4, and about half of the eIF4A-dependent mRNAs fell into our VIC–oligo-4-sensitive group (*Figure 6—figure supplement 2*). However, in comparing our data set with the available information on the genome-wide effects of eIF4A inhibition or downregulation, we find that ES6S blockage seems to have a wider impact on translation than eIF4A downregulation (*Modelska et al., 2015*; *Rubio et al., 2014*; *Wolfe et al., 2014*).

We also tested the effect of the overexpression of two different eS4–EGFP fusion proteins on the accumulation of some representative proteins. Notably, translation of AKT-1 mRNA was strongly inhibited by eS4–EGFP overexpression, whereas the accumulation of GRK2 and HRAS proteins was less affected. A slight increase in eIF4B protein level was also observed upon transfection of eS4–EGFP (*Figure 6d*). This confirmed that ES6S blockage by both VIC–oligo 4 and eS4–EGFP resulted in comparable effects on translation. mRNA threading into the ES6S region makes scanning slower but more processive.

To better understand the influence of the ES6S region on the scanning process, we studied the effect of VIC–oligo 4 on translation in vitro. As observed in transfected cells, the addition of VIC–oligo 4 to rabbit reticulocyte lysates (RRL) drastically inhibited the translation of mRNAs bearing stable RNA stem-loops in their 5′ UTRs. However, for mRNAs with short and unstructured 5′ UTRs (63

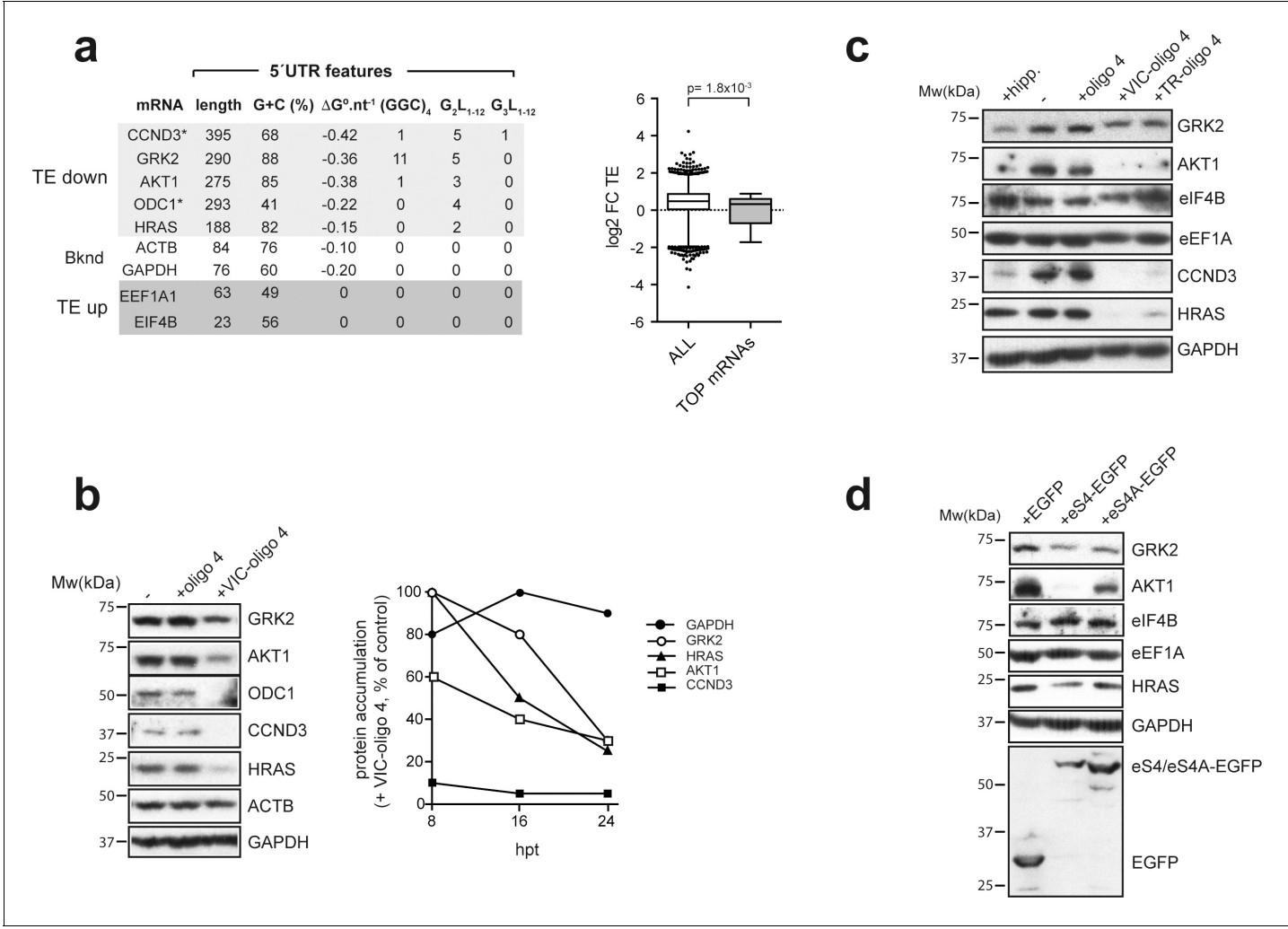

**Figure 6.** Effect of VIC–oligo 4 and TR–oligo 4 on the translation of mRNAs representative of the 'TE down' and 'TE up' groups. (a) 5' UTR features of the representative mRNAs selected for the analysis, with structure stability per nucleotide shown, as well as the presence of (GGC/A)$_4$ and G4s (G$_2$L$_{1-12}$ and G$_3$L$_{1-12}$) (left panel). The right panel shows a comparative analysis of log$_2$ FC TE between all mRNAs and those TOP mRNAs that we detected in our dataset (36). p-value after U test is shown. (b) (Left) Western blot of the accumulated proteins at 24 hr post-transfection (hpt) of the indicated oligos in HEK293T cells. (Right) Time course of protein accumulation at 8 hr, 16 hr and 24 hr post-transfection (hpt) of VIC–oligo 4, with data represented as percentage of the control values (+ oligo 4). (c) Comparative effect of hippuristanol (hipp), VIC–oligo 4 and TR–oligo 4 on the indicated protein levels. Data were analyzed by western blot at 24 hpt with the indicated oligonucleotides; hipp was used at 1 µM. (d) Effect of eS4–EGFP and eS4A–EGFP overexpression on the accumulation of some representative proteins. HEK293T cells were transfected with plasmids expressing the indicated proteins and cell extracts were analyzed by western blot at 48 hpt.

The online version of this article includes the following figure supplement(s) for figure 6:

**Figure supplement 1.** Comparative analysis of the 5' UTR of TOP mRNAs detected in our dataset.

**Figure supplement 2.** Raw and processed data for representative mRNAs and analysis of the functional overlapping between ES6S region and eIF4A.

nt and G-less), translation was unaffected or even stimulated by VIC–oligo 4 when reactions were measured at end-point times (*Figure 7a*). To better understand this, we analyzed the effect of FITC–oligo 4 in real-time translation experiments. Continuous recording of luc activity has been used to measure the time required to detect luc activity (full-translation time, FTT) and its dependence on scanning time (*Vassilenko et al., 2011*). Thus, FTT increases linearly with 5' UTR length, as the 48S-PIC requires more time to complete the scanning of the mRNA when a long 5' is present (*Vassilenko et al., 2011*). Surprisingly, FITC–oligo 4 accelerated the accumulation of luc activity in the reactions programmed with 5' UTR G-less mRNA, and to a lesser extent in those programmed

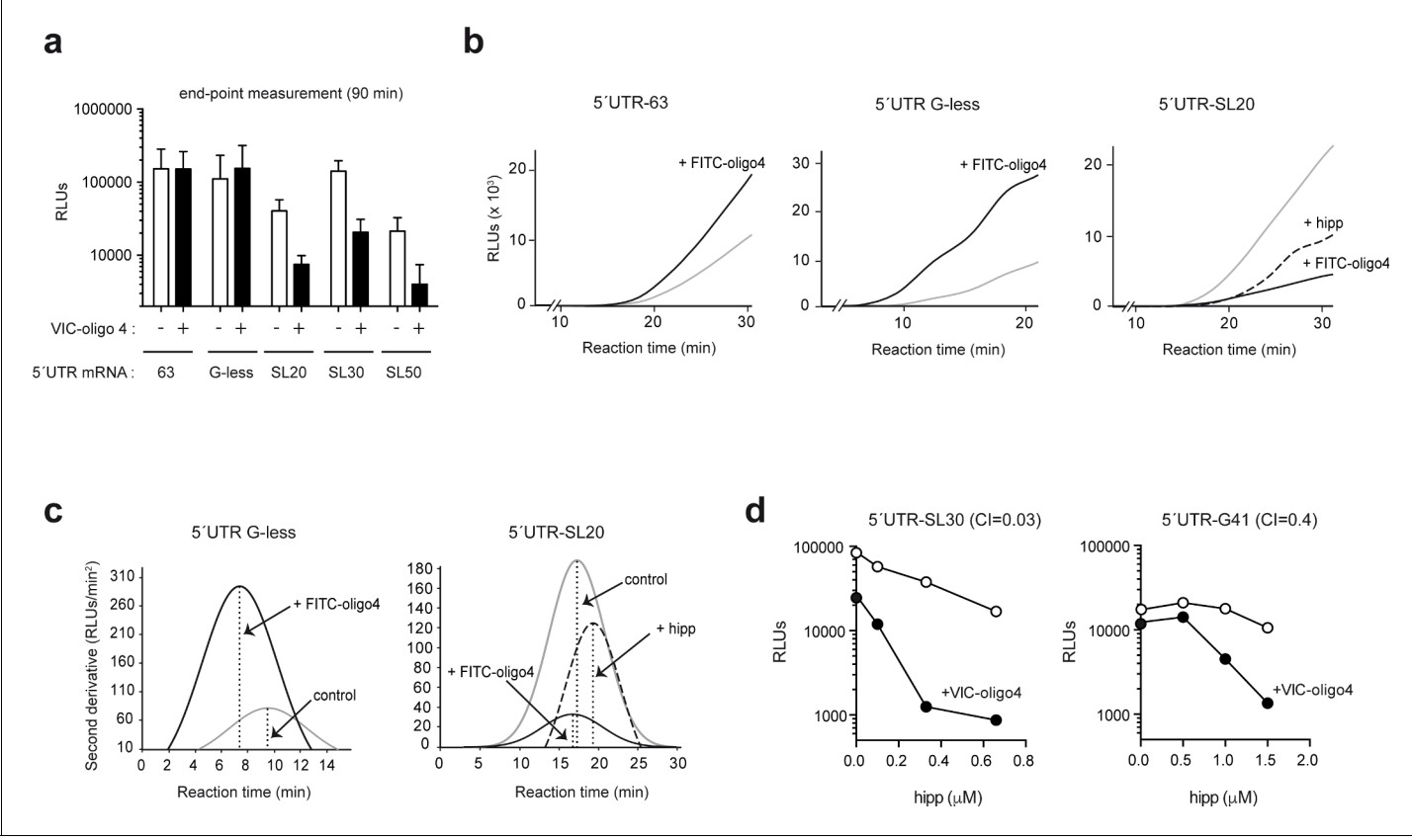

**Figure 7.** mRNA threading into the ES6S region slows down scanning but makes it more processive. (**a**) Effect of FITC-oligo 4 on the translation of Luc mRNAs with different 5' UTRs in RRL. Translation mixtures were incubated for 90 min, which represented the endpoint measurement because no further increase in luc activity was detected. Data are represented as the mean ± SD from at least three independent experiments. (**b**) Luc activity accumulation in continuously recording experiments programmed with the indicated mRNAs. Measurements were taken every 3 min: gray line, no oligo; black line, +FITC–oligo 4. Hipp was added to the indicated samples at a concentration of 2 μM (dashed line). (**c**) Estimates of full translation time (FTT) for 5' UTR G-less and 5' UTR-SL20 mRNAs, and the effect of FITC–oligo 4 and hipp on FTT. Data from panel (B) were processed as described before (*Vassilenko et al., 2011*). The determined FTT values were: 5' UTR G-less = 9.53 min; 5' UTR G-less+FITC–oligo 4 = 7.89 min; 5' UTR-SL20 = 17.57 min; 5' UTR-SL20+FITC–oligo 4 = 17.56 min; 5' UTR-SL20+hipp = 19.44 min. (**d**) Synergistic inhibitory effect of VIC–oligo 4 and hipp on translation of 5'UTR SL30-Luc and 5' UTR G4-1-Luc mRNAs in RRL. Translation mixtures were preincubated with 6 μM of VIC–oligo 4 and with increasing concentrations of hipp for 5 min. Then, mRNAs were added and measurements were taken 90 min later; the calculated combination index (CI) for each mRNA is indicated.

with 5'UTR 63 mRNA (*Figure 7b*). FITC–oligo 4 reduced the FTT of 5' UTR G-less mRNA by 2 min, suggesting accelerated scanning (*Figure 7c*).

In parallel experiments using highly structured 5' UTR-SL20 mRNA, the addition of FITC–oligo 4 drastically reduced the accumulation of luc activity at every time point, although the FTT was not significantly altered, suggesting arrest rather than delay of scanning (*Figure 7c*). Interestingly, addition of hipp to 5' UTR-SL20 mRNA delayed the accumulation of luc activity, which followed a differently shaped curve than that observed for FITC–oligo 4 (*Figure 7b*). In contrast to the addition of FITC–oligo 4, the addition of hipp increased the FTT by 2 min, suggesting a delay in the scanning process. Thus, the effect of eIF4A inhibition was similar but not identical to that observed for FITC–oligo 4 on mRNAs that have stable structures in their 5' UTRs.

To further explore the functional links between the ES6S region and eIF4A, we tested the combined effect of VIC–oligo 4 and hipp on the translation of mRNA with stem-loop (5' UTR-SL30) or G4 (5' UTR-G4-1). To this end, we used suboptimal concentrations of VIC–oligo 4 that induced a moderate blockage of translation (2–3-fold), in combination with increasing concentrations of hipp. Clearly, the combination of VIC–oligo 4 and 0.3 μM hipp induced a dramatic synergistic inhibition of

translation of 5′ UTR-SL30 mRNA (up to 60-fold), whereas the effect on 5′ UTR-G41 translation was less dramatic (12-fold inhibition) (*Figure 7d*). These results strongly suggest that the function of eIF4A helicase and the ES6S region converge during the initiation step.

## Discussion

Here, we present structural, biochemical and functional data supporting a role of the ES6S region in the scanning process, acting as a threading region where mRNA is unwound before entering the classical mRNA channel of the 40S subunit. Rather than a narrow cleft like the decoding groove, the topology of the ES6S region resembles an open channel in which some differences in the accommodation of mRNA molecules are found. The presumed flexibility of ES6S extensions, especially of ES6S$^A$ and ES6S$^B$, may assist in the placement of mRNAs according to their secondary structure content. These extensions are similar to the tentacle-like structures described recently for some ESs of the 60S subunit, which are involved in the recruitment of MetAP and NatA proteins to 80S (*Fujii et al., 2018*; *Knorr et al., 2019*). Moreover, as eIF4G has been found to contact both ES6S$^E$ and ES6S$^B$ extensions (*Yu et al., 2011*), direct participation of eIF4F in the accommodation of the mRNA in the ES6S region seems likely. Three lines of evidence presented here suggest that eIF4A helicase acts on the mRNA in the ES6S region: (1) crosslinking of eIF4A with mRNA was only detected when 4-thio-Us were placed in mRNA positions that fit in the ES6S region of 48S-PIC model; (2) Hipp and VIC–oligo 4 exhibited a strong inhibitory synergy when assayed on 5′ UTR-SL30 mRNA in vitro; and (3) ES6S blockage affected the translation of mRNAs with a 5′ UTR that is enriched in classical dsRNA structures and/or G4s, similar to that previously described when eIF4A activity was inhibited by drugs or silenced (*Modelska et al., 2015*; *Rubio et al., 2014*; *Wolfe et al., 2014*). Nonetheless, the exact placement of the eIF4A helicase bound to mRNA in the scanning complex could not yet be mapped, most probably because of the dynamic nature of eIF4A–mRNA interactions. Besides eIF4A, we also detected interaction of eIF3g with mRNA in the ES6S region, which fits well with its binding to eIF3b very close to the ES6S$^A$ helix (*des Georges et al., 2015*; *Eliseev et al., 2018*). Furthermore, some mutations in the RNA-binding domain of eIF3g have shown decreased processivity of scanning of the structured 5′ UTRs of yeast mRNA (*Cuchalová et al., 2010*). The possibility that other RNA helicases, such as DDX3, DHX9 and DHX36, could also associate with the ES6S region to unwind dsRNA and G4 structures in mRNA deserves further investigation.

The blocking activity showed by oligo 4 was striking because it was only observed when relatively bulky molecules (such as VIC) were conjugated to its 5′ extreme. Although the exact arrangement of VIC–oligo 4 bound to 40S was not determined, the fluorophores probably project toward the cavity of the ES6S region, at least partially blocking it. However, we cannot rule out the possibility that the binding of VIC–oligo 4 to the target sequence may also restrict the conformational changes that ES6S$^A$ and ES6S$^B$ helices have been shown to undergo during initiation (*Melnikov et al., 2012*; *Toribio et al., 2016a*). The differential sensitivity of mRNAs to the translational blockage imposed by VIC–oligo 4 reveals the existence of a great diversity in mRNA-dependence for scanning. Thus, the translation of mRNAs with relatively long and structured 5′ UTRs (like CCND3, H-RAS and ODC-1) was impaired by ES6S blockage, whereas the translation of mRNAs with short and unstructured 5′ UTRs (like many 5′TOP mRNAs) was unaffected or even enhanced by VIC–oligo 4 transfection. This differential effect can be explained if we consider the ES6S region as a true threading path for mRNA that makes the scanning of the 48S-PIC slower but more processive (*Figure 8a*). Thus, translation initiation of mRNAs with short and unstructured 5′ UTRs could be enhanced by blocking of ES6S, as PIC recruitment and scanning of these mRNAs does not require unwinding. Under ES6S blockage, these mRNAs could be threaded directly into the classical mRNA channel, bypassing the ES6S region and thus accelerating the scanning process (*Figure 8b*). In addition, these mRNAs could also better compete for translational machinery in the context of general shut-off induced by VIC–oligo 4. According to this, participation of the ES6S region in ribosomal attachment to mRNA would be dispensable, at least for some mRNAs with unstructured 5′ UTRs such as 5′ TOP mRNAs, which also show little or no requirement for eIF4A (*Gandin et al., 2016*; *Meyuhas and Kahan, 2015*). However, whether mRNA is threaded into the ES6S region during ribosomal attachment, or rather is slotted into the ES6S region once the 5′-cap has entered the classical mRNA channel is still an open question that deserves further investigation.

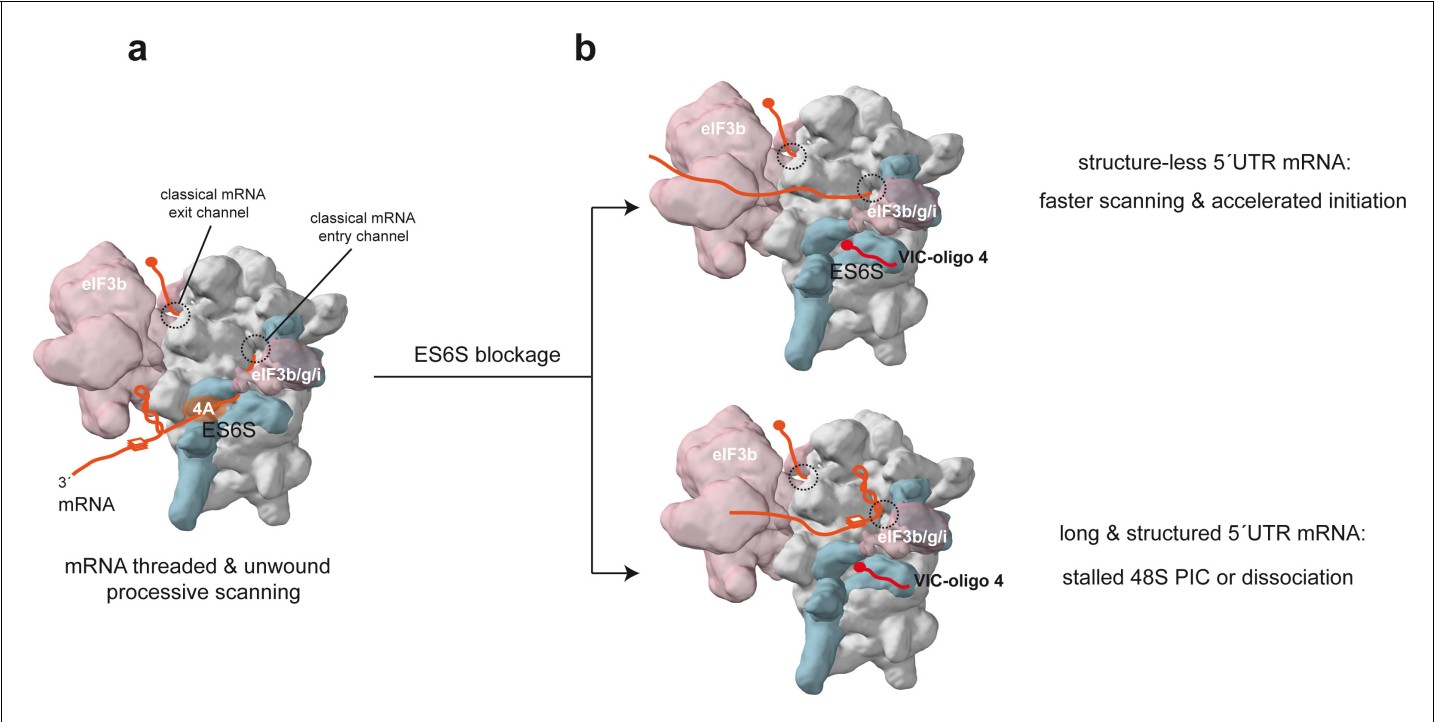

**Figure 8.** Model of the scanning 48S-PIC showing the path of mRNA through the ES6S region. (**a**) Stem-loop (SL) and G4 (three layer square) represent elements of mRNA secondary structure; core eIF3 and some peripheral subunits (3b/3 g/3i) are shown. For simplicity, only the helicase component (eIF4A) of the eIF4F complex is shown. (**b**) Effect of ES6S blockage on mRNA threading. The model represents an extreme situation upon ES6S blockage that excludes mRNA from the ES6S region. For mRNAs with short and unstructured 5′ UTRs, bypassing the ES6S region could accelerate the scanning process (upper). For scanning-dependent mRNAs with long and structured 5′ UTRs, exclusion from the ES6S region would cause the stacking of secondary structural elements at the mRNA entry channel (shown), dissociation of the 48S-PIC, or even the aberrant threading of folded structures into the decoding groove (not shown).

For mRNAs that have long and structured 5′ UTRs, ES6S blockage probably reduces the processivity of the scanning complex along the 5′ UTR. The stem-loop and G4 structures in the 5′ UTR of mRNAs that are not effectively unwound in the ES6S region would become stacked at the mRNA entry channel, resulting in stalling and/or dissociation of the 48S-PIC from the mRNA. Even if the 48S-PIC could bypass some stem-loops without unwinding, as previously observed in vitro (*Abaeva et al., 2011*), it is not clear whether these aberrant complexes would end up being productive. The DHX29 helicase located at the mRNA entry channel of PIC may also alleviate the eventual stacking of some RNA structures at the mRNA entry channel (*Hashem et al., 2013*), although its involvement in the genome-wide unwinding of stable stem-loops and G4s is unknown.

Finally, our data point out the ES6S region as a novel target for small molecules (oligos and aptamers) that could block the translation of specific mRNA subsets, for example those involved in signal transduction and oncogenesis that show extensive RNA structure in their 5′ UTR. This could represent a more specific alternative to the use of eIF4A inhibitors as anti-cancer drugs (*Chu and Pelletier, 2015*).

## Materials and methods

### Oligonucleotides and recombinant DNA

A complete list of the oligonucleotides used in this work can be found in *Supplementary file 2*. Oligonucleotides were purchased from Sigma except for VIC-conjugated versions, which were purchased from Life technologies. In some in vitro experiments using nuclease-treated RRL (Promega), we used a derivative of oligo 4 with phosphorothioate and 2′O-Met modifications that provided increased stability. Templates for in vitro transcription were generated by PCR using

the corresponding oligonucleotides, and the resulting products were purified by the DNA Clean and Concentrator-25 kit (Epigenetics). Firefly luciferase constructs bearing different 5′UTRs were constructed in the pLuc plasmid, derived from pEGFP-N1 (Clontech) by exchanging the EGFP CDS for the luciferase-coding sequence using *BamH*I and *Xba*I sites. The sequences of the 5′UTRs are shown in *Supplementary file 1*.

To construct eS4-EGFP fusion proteins, cDNA of human eS4 mRNA (NM_001007.4) was obtained by reverse transcription of total RNA from HeLa cells using the primer fw-hRPS4X, followed by PCR amplification with primers fw-hRPS4X and rev-hRPS4X. The resulting PCR product was cloned into pEGFP-N1 (Clontech) using *Nhe*I and *Hind*III, resulting in a frame fusion of the two proteins separated by a 23-aa linker (LQDPKLRILQSTVPRARDPPVAT). eS4–EGFP was constructed by cloning the above PCR product using a similar strategy, but the resulting PCR product was cloned into *Nhe*I and *Pst*I sites, resulting in a shorter linker (LQSTATS).

## RNA synthesis and purification

RNAs were in vitro synthesized using HiScribe T7 Quick (NEB) as described previously (*Toribio et al., 2018*). Where indicated, 4-thio-U and/or 30 μCi [α-$^{32}$P]GTP were included in the reaction, and the resulting mRNAs were purified through Chromaspin-30 columns (Clontech). To prepare Luc mRNAs for in vitro translation, mRNA transcripts were capped using the vaccinia capping system (NEB) and purified using the RNA Clean and Concentrator-25 kit (Epigenetics). All of the mRNAs used in this work contained a poly(A) tail of 25 nt.

## Cell culture and transfection

MEFs, HeLa and HEK293T cells were grown in DMEM supplemented with 10% fetal calf serum. Cells were authenticated by microscopic examination and they were free of mycoplasma contamination. For oligonucleotide transfection, cells were grown in 24-well plates at 60–70% confluency and then transfected with 100 pmol of oligonucleotide using TurboFect (Thermo Fisher Scientific) or Lipo-transfectin (NiborLab). Oligonucleotide uptake was monitored using a fluorescence microscope. At the indicated times, cells were fixed for immunofluorescence (IF), infected with the indicated viruses or metabolically labeled with [$^{35}$S]-Met/Cys for 1 hr and analyzed as described previously (*Ventoso et al., 2006*).

## esiRNA-mediated interference

To silence the expression of eIF4A1 gene, HeLa cells ($\approx 5 \times 10^4$ growing in 6-well plates) were transfected with 0.6 μg of esiRNA (EHU-11150–1, SIGMA) targeting human eIF4A1 mRNA (NM_001416), using INTEREFERin (PolyPlus) as facilitator. Three days later, the level of eIF4A1 protein was analyzed by WB.

## Polysome analysis and profiling

Two subconfluent p100 plates of MEF or one plate of HEK293T cells were transfected with the indicated oligonucleotide for 12 hr and lysed in polysome buffer (Tris-HCl 30 mM [pH 7.5], 100 mM KCl, 5 mM MgCl$_2$, 1 mM DTT, 1% Triton X-100 and 50 μg/ml cycloheximide) for 10 min on ice. After three passages through a 22G needle, cell lysates were clarified by low-speed centrifugation and loaded on a 10–40% sucrose gradient prepared in polysome buffer. Gradients were centrifuged at 35,000 rpm in a SW-40 rotor for 3 hr at 4°C in a Beckman SW40.1 rotor and fractionated from the bottom using an ISCO fractionator coupled to a UV recorder. For protein analysis, fractions were extracted with a methanol-chloroform protocol and analyzed by WB with the indicated antibodies. For RNA analysis, fractions were extracted with phenol, ethanol precipitated and pooled in submonosomal (40S, 48S and 60S), monosomal (80S) and polysomal (>2 ribosomes per mRNA) fractions.

## RNA-seq and data processing

A total of 12 samples were prepared for sequencing. To track the dilutions made during the construction of RNA-seq libraries, 0.3 μL of 50X Mix 1 dilution of ERCC RNA spike-in mix (Thermo Fisher Scientific Fisher) was added to the samples prior to rRNA depletion. The cDNA libraries were prepared with 1 μg of rRNA-depleted RNA using the TruSeq Stranded mRNA Sample Preparation kit (Illumina, Inc). RNA-seq was carried out on an Illumina HiSeq 2500 platform, and raw data quality

control was generated using FastQC. We used the GRCh38 human genome release and Ensembl annotation for further analysis. Bowtie 1.1.2 (*Langmead et al., 2009*) was used with default parameters to remove rRNA reads, to estimate the differential dilution of the samples via the spike-in, and to align RNA reads. About 75% of reads were aligned in each sample. Alignments were refined with Hisat2 2.1.0 (*Kim et al., 2015*) using the *–known-splicesite-infile* parameter to define the intron limits. Read counting per gene was done with Featurecounts release 1.6.3 (*Liao et al., 2014*). The library size and spike-in were used to normalize counts using the median of ratios method (*Love et al., 2014*). The DESeq2 R package was used to estimate the translation efficiency (TE) of each sample and to compare TE between samples (*Love et al., 2014*). We defined TE as the ratio of the mRNA abundance in the polysomal fraction divided by the abundance in the monosomal fraction. Change in TE was calculated as $TE_{oligo\ 4}/TE_{VIC-oligo4}$ and expressed in $\log_2$fold change TE (FC TE). To generate the database with feature annotations for every mRNA, we integrated information from the Ensembl human 95 database with stable RefSeq ID, with Transcript Support Level (TSL) and APRIS annotation. We also incorporated experimental mapping of the +one transcription start site (TSSs) using the nanocage technique that was available for 2500 mRNAs of HEK293T cells (*Gandin et al., 2016*). RNA secondary structures were predicted using the RNAfold 2.2.10 algorithm of the ViennaRNA package (*Lorenz et al., 2011*). For the analysis, the minimum folding energy of the centroid structure was used, including G4s or not.

## UV crosslinking experiments

Crosslinking experiments using [$^{32}$P]-labeled mRNAs with photoreactive 4-thio-U were carried out as previously described (*Toribio et al., 2016a*). 48S or 80S complexes were assembled with GMP-PNP or cycloheximide, respectively, and incubated at 30°C for 20 min. After crosslinking at 360 nm, lysates were centrifuged over a 20% sucrose cushion at 90,000 x g for 3 hr, and the whole ribosomal fraction (WRF) was resuspended in 50 μL of TE buffer. For protein analysis, samples were digested with RNAse A and T1 for 1 hr at 37°C before SDS-PAGE analysis as described previously (*Toribio et al., 2016a*).

## Mapping mRNA–18S rRNA contacts

To identify contacts between mRNA and 18S rRNA within the 48S PIC, a large-scale experiment using 0.5 ml of RRL and cold 4-thio-U labeled mRNAs was carried out. After UV crosslinking at 360 nm, the WRF was denatured in 500 μl buffer D (Tris-HCl 30 mM [pH 7.5], 0.5 M LiCl, 0.5% LiDS, 0.5 mM EDTA and 1 mM DTT) and poly(A)+ mRNA was captured with oligo(dT) magnetic beads (NEB) under denaturing conditions. After extensive washing, RNA was eluted with 100 μl $H_2O$ and concentrated by ethanol precipitation in the presence of glycogen. Captured 18S RNA that was crosslinked to mRNA was analyzed by reverse transcriptase termination site (RTTS) assays as described previously (*Kielpinski et al., 2013*). Briefly, 18S rRNA was retrotranscribed with Superscript IV (Invitrogen) using 2 pmol of primer 3 followed by a 20 min digestion with RNAse H. The resulting cDNA population was purified with AMPURE beads and 3′ ligated to primer 3′ RTTS_adapter with CirLigase (Illumina). Then, a 30-cycle PCR with oligos 3′_adapter and RP_FWD was carried out, followed by a final 10-cycle PCR with oligo RP_FWD and oligos RP_REV_INDEX 5, 6, 7 or 12 (see *Supplementary file 1*). The resulting libraries were sequenced in a MiSeq system (Illumina), and reads were aligned to rabbit 18S rRNA with Bowtie 2 using the default parameters. The crosslinking sites in 18S were identified as the 5′-adjacent nucleotide to the 5′ end of every aligned read.

## Mapping mRNA–protein contacts

To identify proteins that crosslinked to unlabeled mRNA in the 48S-PIC, a large-scale experiment using 0.5 ml of RRL and 3 μg of the indicated mRNA were used. After 30 min crosslinking at 254 nm, WRFs were resuspended in buffer D (Tris-HCl 30 mM [pH 7.5], 0.5 M LiCl, 0.5% LiDS, 0.5 mM EDTA and 1 mM DTT) and poly(A)+ mRNA was captured with oligo(dT) magnetic beads (NEB) under denaturing conditions as described above. After extensive washing, RNA was eluted with 100 μl $H_2O$ and digested with RNAse A and T1. Samples were trypsin digested, concentrated and analyzed by high-resolution LC-ESI-MS/MS.

## Denaturing immunoprecipitation

dIP was carried out as described recently (*Toribio et al., 2018*). Briefly, the ribosomal fraction from a 500 µl translation reaction including $2 \times 10^6$ cpms of [$\alpha$-$^{32}$P]−4-thio-U-SV-DLP U1 mRNA and 2 mM GMP-PNP was obtained as described previously (*Toribio et al., 2016a*). The WRF was obtained as above and denatured in a buffer containing Tris-HCl 50 mM (pH 7.4), 5 mM EDTA, 10 mM DTT and 5% w/v SDS. After 5 min boiling, samples were kept on ice and slowly renatured in Tris-HCl 50 mM (pH 7.4), 150 mM NaCl, 5 mM EDTA, and 1% Triton X-100. Then, the extract was split and incubated with the indicated antibodies overnight at 4°C. Next, a mixture of protein A/G conjugated to magnetic beads (Thermo Fisher Scientific) was added and incubated for 1 hr at room temperature, and immunoprecipitated complexes were analyzed by SDS-PAGE and autoradiography.

## Mapping of 3′ oligo 4–18S rRNA pairing

5 pmol of oligo 4 or VIC–oligo 4 were incubated with 30 pmol of 40S subunits purified from RRL and incubated for 30 min at 30°C in polysome buffer. Then, DNA–RNA hybrids were digested with 5 U of RNAse H (NEB) for 15 min at 37°C, extracted with phenol and ethanol precipitated. The 3′ end of the resulting RNA fragments was polyadenylated with poly(A) polymerase (NEB) and retrotranscribed with oligo(dT) primer. Finally, the resulting cDNA was amplified by PCR using oligo(dT) and oligo 9 primers, and sequenced. Data confirmed that the final 6 nt at the 3′ end of oligo 4 remained unpaired, and that the first nt of 18S rRNA that paired with the oligo 4 was C835.

## Western blot and immunofluorescence

Western blots were carried out as described previously (*Ventoso et al., 2006*) using the following primary antibodies: anti-RPS4X (sc-85133, Santa Cruz Biotech.), anti-RPS7 (sc-377317, Santa Cruz Biotech.), anti-RPS6 (sc-74576, Santa Cruz Biotech.), anti-eIF4A (STJ2724, St. John´s lab,), anti-eIF3g (STJ23512, St. John´s lab), anti-EGFP (11814460001, Roche), anti-eEF1A1 (2551, Cell Signaling), anti-AKT1 (9272, Cell Signaling), anti-ACTB (T-5168, Sigma), anti-eIF4b (sc-376062, Santa Cruz Biotech.), anti-HRas (sc-53959, Santa Cruz Biotech.), anti-CCND3 (sc-453, Santa Cruz Biotech.), anti-ODC1 (sc-398116, Santa Cruz Biotech.), anti-GRK2 (a gift from C. Murga, CBMSO). Blots were developed with ECL (GE) and bands were quantified by densitometry. Immunofluorescence analysis was carried out as described previously using anti-RPS7 (1:500) and anti-mouse Alexa 595 as secondary antibody (Invitrogen). The preparations were analyzed under a Nikon A1R confocal microscope.

## Protein and oligonucleotide modeling

Models of the human PIC (EMD-5658), 80S (EMD-5326) and rabbit 48S PIC (PDB: 4KZZ) were visualized in Chimera (*Pettersen et al., 2004*). Models of eS4–EGFP were generated in Phyre2 (*Kelley et al., 2015*) and I-Tasser (*Yang et al., 2015*). Both programs rendered models covering 95% of protein length at high confidence (>90%), showing a root mean square deviation (RMSD) of 0.5–1 Å. The linker was predicted to adopt a random coil conformation, so the resulting models differed somewhat in the relative orientation of the two proteins. eS4–EGFP was modeled into the human 40S subunit (EMD-5326) using the matchmaker command of Chimera with default parameters. Models showing clashes between EGFP and ES6SC-D were discarded.

## In vitro translation and luciferase measurement

Translations were carried out in 10–15 µl samples containing 70% vol of RRL (Promega) and 50 ng of luciferase mRNA, and reactions were incubated at 30°C for the indicated times. For continuous recording of luc activity, 1 µl samples were taken every 2 min and kept on ice. Luc activity was measured on a Berthold luminometer.

## Acknowledgements

We are indebted to Jerry Pelletier (McGill University) for providing us with hippuristanol, Cristina Murga (CBMSO, Madrid) for the GRK2 antibody and Fernando Rodriguez Pascual for INTERFERin. We also thank the Proteomic Facility at the Centro Nacional de Biotecnología (CSIC, Madrid) for protein identification by mass spectrometry analysis. The technical support of Laura Barbado is also acknowledged.

## Additional information

### Funding

| Funder | Grant reference number | Author |
|---|---|---|
| Ministerio de Economía y Competitividad | BFU2013-45003-R | Iván Ventoso |
| Ministerio de Economía y Competitividad | BFU2017-84955-R | Iván Ventoso |

The funders had no role in study design, data collection and interpretation, or the decision to submit the work for publication.

### Author contributions

Irene Díaz-López, Data curation, Formal analysis, Validation, Investigation, Visualization, Methodology; René Toribio, Conceptualization, Data curation, Formal analysis, Supervision, Validation, Investigation, Methodology; Juan José Berlanga, Investigation, Methodology; Iván Ventoso, Conceptualization, Resources, Supervision, Funding acquisition, Validation, Investigation, Writing—original draft, Writing—review and editing

### Author ORCIDs

Juan José Berlanga (iD) http://orcid.org/0000-0002-2408-6561
Iván Ventoso (iD) https://orcid.org/0000-0001-7887-3520

### Decision letter and Author response

Decision letter https://doi.org/10.7554/eLife.48246.sa1
Author response https://doi.org/10.7554/eLife.48246.sa2

## Additional files

### Supplementary files

• Supplementary file 1. Sequence and secondary structure of the 5′ UTRs used in study.The predicted secondary structure (RNAfold) is depicted in dot-bracket notation. Stem-loops are marked in red together with the predicted stability ($\Delta G°$). The predicted stability for the entire 5′ UTR is shown, including the correction for 5′ UTR length. For the 5′ UTR of luc mRNAs, shared sequences are in bold.

• Supplementary file 2. Table of primers.The sequences of oligonucleotides used for blocking the ES6S region and for PCR amplification are shown.

• Transparent reporting form

### Data availability

GEO accession number: GSE129651.

The following dataset was generated:

| Author(s) | Year | Dataset title | Dataset URL | Database and Identifier |
|---|---|---|---|---|
| Irene Díaz-López, René Toribio, Juan José Berlanga, Iván Ventoso | 2019 | GSE129651 | https://www.ncbi.nlm.nih.gov/geo/query/acc.cgi?acc=GSE129651 | NCBI Gene Expression Omnibus, GSE129651 |

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
