## [Decision Letter]

**Acceptance summary:**

The paper provides a strong mechanistic insight into the mechanism that explains the inverse correlation between mRNA 5'UTR G+C content/ ΔG and mRNA translation efficiency. It bolsters the model for the function of eIF4A in unwinding the mRNA 5' secondary structure to facilitate ribosome recruitment. They have documented the role of the ribosomal ES6S region in the control of ribosome scanning. They provide evidence that mRNA is threaded into the ES6S region of 48S pre-initiation complex (PIC), and unwound by the RNA helicase eIF4AI, rendering the 48S PIC more processive, particularly on mRNAs with long and structured 5´UTRs. It would be important in the future to distinguish between the activities of eIF4AI and eIF4AII because the functions of eIF4AI and eIF4AII are were suggested in some studies not to be equivalent.

**Decision letter after peer review:**

Thank you for submitting your article "An mRNA-threading channel in the ES6S region of the translation 48S-PIC promotes RNA unwinding and scanning" for consideration by *eLife*. Your article has been reviewed by three peer reviewers, including Nahum Sonenberg as the Reviewing Editor and Reviewer #1, and the evaluation has been overseen by a Reviewing Editor and James Manley as the Senior Editor.

The reviewers have discussed the reviews with one another and the Reviewing Editor has drafted this decision to help you prepare a revised submission.

Summary:

This is an interesting manuscript that explores the role of a specific topological region of the ribosome in translation initiation. The authors performed experiments to interfere with the ES6S region of the 48S-PIC and report that this selectively blocks translation of mRNAs with structured 5' UTRs. They interpret their data to indicate that mRNAs that thread into this region are scanned slower and in a more processive manner than mRNAs that bypass this region. Overall this is a good study, but there are some weaknesses, including missing controls and interpretation of the experiments. The reviewers raised many important points, that you can see in the full reviews. The concerns listed below are the most critical.

Essential revisions:

1) Consistent with a previous study, the authors show that 4-thio-U crosslinking identifies an interaction between unstructured and viral mRNA with eIF3g and eIF4A. However, only eIF3g is found to crosslink to this region on the unstructured mRNA, raising a question about how general the position of eIF4A may be on non-viral mRNAs.

2) A major weakness of the study is that the authors have not directly shown whether the oligo-fluorophore conjugate that is complementary to ES6Sd and EGFP-eS4 actually inhibits the interaction of the mRNA and the ES6S region. All interpretations of the experiments assume this, but this should be directly tested using the 4-thio-U crosslinking experiment in the presence of the oligo and EGFP-eS4.

3) The authors monitor the rate of luciferase translation in the presence of oligo and hippuristanol in Figure 7. The delay in appearance of LUC protein and the synergistic inhibition is interpreted to mean that scanning rates are slower. This seems a bit of a strong statement for an assay that only monitors the rate of luciferase translation. Without additional evidence that more rigorously tests the rate of "scanning", the authors should significantly tone down the interpretation of this experiment (that mRNA threading into the ES6S region makes scanning slower but more processive).

The evidence for mRNA "threading" model in this study is not very strong. The claim that eIF4A may function in the ES6S region to unwind duplex regions should be qualified, as the eIF4A crosslink is only studied in the presence of viral mRNAs.

4) The appropriate control for Figure 3—figure supplement 2B would be VIC-oligo C.

5) Does GFP tagging of another ribosomal protein yield the same results as eS4-EGFP?

6) Figure 4A. How do the mRNAs, including HCV IRES respond to VIC-oligo C (the control)? This negative control is missing. The sequences of the 5' UTRs of the reporters used here are shown as some computer printout in Supplementary file 1 in a few crude ways with no attempt to make this reader-friendly – this should be corrected. Please confirm that there are no upstream AUGs in any of the reporters used, as this could confound interpretations (by my visual scan there didn't appear to be).

7) The authors do not specify whether the antibody they used for IP in the cross-linking experiments is anti-4AI or 4AII. This was neither done in their previous paper (Toribio et al., 2018). Because the functions of eIF4AI and eIF4AII are not equivalent, it is important to show which eIF4A is studied in this paper, and some discussion about the different functions of 4AI and II would be very informative to the reader.

*Reviewer #1:*

The paper provides a strong mechanistic insight towards the understanding of the inverse correlation between mRNA 5'UTR G+C content/ ΔG and mRNA translation efficiency. It provides further support for the function of eIF4A in unwinding the mRNA 5' secondary structure to facilitate ribosome recruitment. However, it is surprising that the authors have not made use of their excellent experimental system to distinguish between the activities of eIF4AI and eIF4AII. In fact, they do not specify whether the antibody they used for IP in the cross-linking experiments is anti-4AI or 4AII. This was neither done in their previous paper (Toribio et al., 2018). Because the functions of eIF4AI and eIF4AII are not equivalent, it is important to show which eIF4A is studied in this paper, and some discussion about the different functions of 4AI and II would be very informative to the reader.

Figure 4B; It is not immediately clear why the translation of EMCV and poliovirus RNAs is not affected by VIC-Oligo4, inasmuch as the translation of both is eIF4AI dependent.

Figure 5A; The ribosome profile in the polysome region appears abnormal. The experiment must have been repeated, but I could not find how many times this experiment (and several others) was repeated. Could they show a different profile?

*Reviewer #2:*

In this manuscript, Ventoso and colleagues undertake experiments to interfere with the ES6S region of the 48S-PIC and report that this selectively blocks translation of mRNAs with structured 5' UTRs. They interpret their data to indicate that mRNAs that thread into this region are scanned slower and in a more processive manner than mRNAs that bypass this region. It's an interesting manuscript that explores the role of a specific topological region of the ribosome in translation initiation.

Essential revisions:

- I believe an appropriate control for Figure 3—figure supplement 2B would be VIC-oligo C.

- Does GFP tagging of another ribosomal protein yield the same results as eS4-EGFP?

- Shouldn't there be a length control for SL20 in Figure 4A? This is also a good opportunity for the authors to test how systematically increasing structure tracks with translational efficiency using this assay. As well, the analysis of the effect of length on efficiency should be more systematically analyzed here. As it stands, the authors have not taken full advantage of the potential of this assay to strengthen their cause.

- Figure 4A. How do these mRNAs, including HCV IRES respond to VIC-oligo C (the control)? This negative control is missing. The sequences of the 5' UTRs of the reporters used here are shown as some computer printout in Supplementary file 1 in a few crude way with no attempts to make this reader-friendly – this should be corrected. Please confirm that there are no upstream AUGs in any of the reporters used, as this could confound interpretations (by my visual scan there didn't appear to be).

- Figure 4B – SFV. The panel is not properly mounted. Lane 4 seems to have been prematurely cropped on the right and the top panel is not lined up properly with the bottom ethidium stained control.

Figure 5E – Are these values standardized to an internal control? How do these reporters respond to VCI-oligo C (negative control).

- I am sure that the authors know that the Western blots shown in Figure 6 are not reporting on a translational response. Please show distribution of the test mRNAs across polysome gradients or perform metabolic labelled followed by IP to report on nascent protein production.

- Subsection “Genome-wide analysis reveals the role of the ES6S region in translation of mRNAs with G-rich 5´UTRs”. The Rubio and Wolfe studies using rocaglates probably should not be cited here since these cause clamping of 4A to RNA, hence they do not lead to only a loss of 4A activity since the clamped complexes themselves can interfere with translation (see PMID 27309803).

- Do all constructs yield the same final levels of Luciferase in the FTT assay? Why is the area under the curves in Figure 7c between the two conditions very different? Is this reflective of more/less mRNA being translated? Where is the FITC-oligo C control?

- I am not sure how to interpret Figure 7D. The authors indicate that they utilize suboptimal concentrations of VIC-oligo-4 to induce a moderate block of translation (2-3 fold). Since to get an effect of VIC-oligo4 you only need one oligo bound to the ribosomal target, it means there is a pool of ribosomes that are also not bound by the oligo. I am not sure what exactly the "synergy obtained with hipp means – does it mean synergistic effects with those ribosomes having a bound VIC-oligo-4, or does it mean hipp is acting on those ribosomes not having a VIC-oligo 4 bound and exerting an effect unrelated to the ES6S region and this is being "scored" as a synergistic effect. Where are the controls with VIC-oligo C.

- The authors refer to the synthetic 128-nt mRNA as "flat". This is somewhat vernacular and the authors might want to change this to a more scientifically appropriate label.

- Subsection “The path of mRNA through the ES6S region of the 48S PIC”. The authors write "According to our data, eIF4A may be placed a bit further downstream, probably between the ES6SA 179 and ES6SB 180 helices (Figure 2C)." I don't see in Figure 2c, where the ES6SA and ES6SB sites are indicated.

*Reviewer #3:*

The authors have two major findings that could potentially transform our understanding of mRNA recruitment.

Firstly, the authors have shown that 4-thio-U crosslinking of the +24 to +34 region of an mRNA occurs around the ES6S region of the 40S subunit (within the 48S PIC). This is obtained using a short unstructured mRNA and GMP-PNP. Consistent with this, non-specific UV crosslinking also reveals interactions between the mRNA and eS4/uS4, which is generally consistent with the ES6S interaction region. Consistent with a previous study, the authors show that 4-thio-U crosslinking identifies an interaction between unstructured and viral mRNA with eIF3g and eIF4A. This crosslinking occurs in the same mRNA region that crosslinks to the ES6S region suggesting that eIF3g and eIF4A are located in this area in the 48S PIC. However, only eIF3g is found to crosslink to this region on the unstructured mRNA, raising a question about how general the position of eIF4A may be on non-viral mRNAs.

The second major finding is that perturbation of the ES6S region, either by an oligo-fluorophore conjugate that is complementary to ES6Sd or by an EGFP fused eS4 protein (located near ES6S), leads to reduced translation rates. This is shown using reporter mRNAs as well as on a genome wide scale. Interestingly, inhibition of translation is evident on mRNAs harboring highly structured and/or long 5'UTRs (including G quadruplex containing 5'UTRs). The specificity of inhibition is controlled for by using the same oligo without the fluorophore or to an oligo that is complementary to a different rRNA region.

The main weakness of the study is that the authors have not directly shown whether the oligo conjugate and EGFP-eS4 actually inhibits the interaction of the mRNA and the ES6S region. All interpretations of the experiments assume this, but this should be directly tested using the 4-thio-U crosslinking experiment in the presence of the oligo and EGFP-eS4.

The authors monitor rate of luciferase translation in the presence of oligo and hippuristanol in Figure 7. The delay in appearance of LUC protein and the synergistic inhibition is interpreted to mean that scanning rates are slower. This seems a bit of a strong statement for an assay that only monitors the rate of luciferase translation. Without additional evidence that more rigorously tests the rate of "scanning", the authors should significantly tone down the interpretation of this experiment (that mRNA threading into the ES6S region makes scanning slower but more processive).

This reviewer doesn't really see strong evidence for mRNA "threading" model from this study. The claim that eIF4A may function at the ES6S region to unwind duplex regions also seems over interpreted – as far as I can see, the eIF4A crosslink only seems to occur in the presence of viral mRNAs.

Overall this is an interesting study, but some major weaknesses with interpretation of the experiments does lower enthusiasm somewhat.

[Editors' note: further revisions were requested prior to acceptance, as described below.]

Thank you for resubmitting your work entitled "An mRNA-binding channel in the ES6S region of the translation 48S-PIC promotes RNA unwinding and scanning" for further consideration at *eLife*. Your revised article has been favorably evaluated by James Manley (Senior Editor), a Reviewing Editor, and one reviewer.

The manuscript has been improved but there are some remaining issues that need to be addressed before acceptance, as outlined below: There is one concern that was not adequately addressed. The authors state in their rebuttal that the St. John lab antibody is against eIF4AI according to the company's datasheet.

"The antibody used here and in previous papers to detect eIF4A by IP experiments was STJ27247 from St. John's lab. According to the datasheet provided by the supplier, the antibody was raised in rabbits using the amino acids 1-406 aa of eIF4AI as immunogen, so that we assume it is specific for eIF4AI. A single protein band was detected by western blot using this antibody, suggesting that the antibody does not react with eIF4AII."

However, the authors need to demonstrate that the antibody is specific to eIF4AI. Also, this information is not presented in the material and methods section and they keep refereeing to eIF4A throughout the paper.

It is important to address this issue because the existing confusing literature that claims that eIF4AII exhibits an antagonistic function to eIF4AI muddles the literature. The authors should alert the readers to this.

---

## [Author Response]

Reviewer #1:The paper provides a strong mechanistic insight towards the understanding of the inverse correlation between mRNA 5'UTR G+C content/ ΔG and mRNA translation efficiency […] This was neither done in their previous paper (Toribio et al., 2018). Because the functions of eIF4AI and eIF4AII are not equivalent, it is important to show which eIF4A is studied in this paper, and some discussion about the different functions of 4AI and II would be very informative to the reader.

The antibody used here and in previous papers to detect eIF4A by IP experiments was STJ27247 from St. John´s lab. According to the data sheet provided by the supplier, the antibody was raised in rabbits using the amino acids 1-406 aa of eIF4AI as immunogen, so that we assume it is specific for eIF4AI. A single protein band was detected by western blot using this antibody, suggesting that the antibody does not react with eIF4AII.

We agree with the reviewer that our system would be an excellent opportunity to test in the future whether eIF4AII could also bind ES6S region of 48S-PIC, a finding that could provide further insights into the mechanism of miRNA-mediated translational arrest.

Figure 4B; It is not immediately clear why the translation of EMCV and poliovirus RNAs is not affected by VIC-Oligo4, inasmuch as the translation of both is eIF4AI dependent.

Our data consistently showed that translation driven by EMCV and PV IRES was unaffected, or only marginally affected by VIC-oligo 4. As the reviewer points out, eIF4AI binds the EMCV and PV IRES (together with eIF4G and other proteins) to actively prepare the mRNA for 43S-PIC attachment. Importantly, this "activation" step occurs before ribosomal attachment so that it would not be affected by the presence of VIC-oligo 4. Since EMC and PV mRNAs do not require scanning (or a minimal scanning for PV mRNA), translation of these mRNAs would be largely independent of ES6S-associated unwinding activity of eIF4A that promotes scanning.

Figure 5A; The ribosome profile in the polysome region appears abnormal. The experiment must have been repeated, but I could not find how many times this experiment (and several others) was repeated. Could they show a different profile?

Figure 5A shows the polysome profile corresponding to exp #1 intended to RNAseq. analysis. This profile can be considered as representative of what we routinely found in HEK293T cells using our protocols. The presence of intact large polysomes in heavy fractions was confirmed by WB with anti-S6 antibodies (lower panel Figure 5—figure supplement 1A). This polysome profile is very similar to the one shown in Figure 3B,

Reviewer #2:[…] Essential revisions:- I believe an appropriate control for Figure 3—figure supplement 2B would be VIC-oligo C.

Wherever possible, we included more than one control oligo in our experiments. The effect of VIC-oligo C was extensively tested, giving similar results to oligo 4 alone (no effect). A new panel (c) in Figure 3—figure supplement 3 comparing the effect of oligo 4, VIC-oligo C and VIC-oligo 4 on translation in MEFs is now included.

- Does GFP tagging of another ribosomal protein yield the same results as eS4-EGFP?

We initially selected two ribosomal proteins surrounding the ES6S region as candidates for GFP tagging: eS4 and eS7. Unfortunately, eS7-GFP fusion accumulated at very low levels in transfected cells and it hardly incorporated into the 40S subunit (results not shown). Due to design limitations, we have not explored any further RPSs for GFP tagging.However, tagging RPSs does not necessarily affect translation as demonstrated using the Halo Tag technology for tagging RACK1 or RPS9 with a protein similar in mass to EGFP (Gallo et al., 2011).

- Shouldn't there be a length control for SL20 in Figure 4A? This is also a good opportunity for the authors to test how systematically increasing structure tracks with translational efficiency using this assay. As well, the analysis of the effect of length on efficiency should be more systematically analyzed here. As it stands, the authors have not taken full advantage of the potential of this assay to strengthen their cause.

The 5´UTR-85 or G-less mRNAs are similar in length to SL20 and they could be used as length controls. We have constructed two additional mRNAs bearing 5´UTR of intermediate length (242 and 427 nts). These mRNAs, together with those bearing 5´UTR of 33 and 656 nts, were tested in a new experiment similar to that described in Figure 4A. As new panel (a) of Figure 4—figure supplement 1 shows, the sensitivity to VIC-oligo 4-mediated translational block increased with 5´UTR length of mRNA as expected.

- Figure 4A. How do these mRNAs, including HCV IRES respond to VIC-oligo C (the control)? This negative control is missing. The sequences of the 5' UTRs of the reporters used here are shown as some computer printout in Supplementary file 1 in a few crude way with no attempts to make this reader-friendly – this should be corrected. Please confirm that there are no upstream AUGs in any of the reporters used, as this could confound interpretations (by my visual scan there didn't appear to be).

We have now included the effect of VIC-oligo C on translation of some reporter mRNAs, including the HCV IRES. The new information is in panel (b) of Figure 4—figure supplement 1.

Regarding the information provided in Supplemental file 1, we confirm that none of the 5´UTRs used here contain upstream AUGs. The sequences are displayed in the output style from Vienna RNA fold server (courier font) that we used for RNA folding predictions. This is an appropriate plain text to line up sequences according to their length.

- Figure 4B – SFV. The panel is not properly mounted. Lane 4 seems to have been prematurely cropped on the right and the top panel is not lined up properly with the bottom ethidium stained control.

The reviewer is right, and the panel has now been properly lined up.

Figure 5E – Are these values standardized to an internal control? How do these reporters respond to VCI-oligo C (negative control).

In these experiments, RRL were programmed with the indicated reporter mRNA alone or in the presence of oligo 4 or VIC-oligo 4. Oligo 4 did not significantly affect translation and these numbers were taken as control. Given the high degree of reproducibility of the results, we did not consider it necessary to include an internal control mRNA. We did not include the VIC-oligo C as control in these experiments.

- I am sure that the authors know that the Western blots shown in Figure 6 are not reporting on a translational response. Please show distribution of the test mRNAs across polysome gradients or perform metabolic labelled followed by IP to report on nascent protein production.

The reviewer is right that western blots does not necessarily report a differential translational response, so that we also included the distribution of representative mRNAs in the monosomal and polysomal fractions (expressed as translation efficiency (TE; poly/mono), together with the changes in "total" mRNA (RNA) in panel (a) of Figure 6—figure supplement 2. As can be seen, these data were consistent with those shown in Figure 6, so that in this case the western blot can be considered as an appropriate read-out of differential translational response.

- Subsection “Genome-wide analysis reveals the role of the ES6S region in translation of mRNAs with G-rich 5´UTRs”. The Rubio and Wolfe studies using rocaglates probably should not be cited here since these cause clamping of 4A to RNA, hence they do not lead to only a loss of 4A activity since the clamped complexes themselves can interfere with translation (see PMID 27309803).

The cited references (Rubio et al., 2014 and Wolf et al., 2014) used silvestrol, but not rocaglates.

- Do all constructs yield the same final levels of Luciferase in the FTT assay? Why is the area under the curves in Figure 7c between the two conditions very different? Is this reflective of more/less mRNA being translated? Where is the FITC-oligo C control?

No, the final levels of luciferase activity varied among the constructs tested as expected. Thus, 5´UTR G-less yielded higher levels of luciferase activity than 5´UTR SL20 as it is shown in panel a of Figure 7. As the reviewer points out, the area under the curves reflects the differences in translation efficiency among mRNAs.

- I am not sure how to interpret Figure 7D. The authors indicate that they utilize suboptimal concentrations of VIC-oligo-4 to induce a moderate block of translation (2-3 fold). Since to get an effect of VIC-oligo4 you only need one oligo bound to the ribosomal target, it means there is a pool of ribosomes that are also not bound by the oligo. I am not sure what exactly the "synergy obtained with hipp means – does it mean synergistic effects with those ribosomes having a bound VIC-oligo-4, or does it mean hipp is acting on those ribosomes not having a VIC-oligo 4 bound and exerting an effect unrelated to the ES6S region and this is being "scored" as a synergistic effect. Where are the controls with VIC-oligo C.

We empirically titrated the amount of VIC-oligo 4 to find a concentration that induced a moderate block (less than three-fold) in translation. Although the binding of VIC-oligo 4 to ribosome should follow a 1:1 stoichiometry, we now know that in our in vitro assays not all the ribosomes were bound to VIC-oligo 4 even at saturating concentration of this oligo. When we combined VIC-oligo 4 and hipp, the dramatic synergistic effect found obviously corresponded to those 48S PIC having bound VIC-oligo and hipp. If not, a simple additive (but not synergistic) effect would be expected. However, we cannot rule out that some of this additive effect can also contribute to the final effect of VIC-oligo 4+ hipp.

-The authors refer to the synthetic 128-nt mRNA as "flat". This is somewhat vernacular and the authors might want to change this to a more scientifically appropriate label.

The reviewer is right that "flat" sounds a bit atypical, so that we have changed it for "unstructured" throughout the text.

- Subsection “The path of mRNA through the ES6S region of the 48S PIC”. The authors write "According to our data, eIF4A may be placed a bit further downstream, probably between the ES6SA 179 and ES6SB 180 helices (Figure 2C)." I don't see in Figure 2C, where the ES6SA and ES6SB sites are indicated.

The positions of ES6SA and ES6SB helices are now indicated in Figure 2C.

Reviewer #3:The authors have two major findings that could potentially transform our understanding of mRNA recruitment.Firstly, the authors have shown that 4-thio-U crosslinking of the +24 to +34 region of an mRNA occurs around the ES6S region of the 40S subunit (within the 48S PIC). This is obtained using a short unstructured mRNA and GMP-PNP. Consistent with this, non-specific UV crosslinking also reveals interactions between the mRNA and eS4/uS4, which is generally consistent with the ES6S interaction region. Consistent with a previous study, the authors show that 4-thio-U crosslinking identifies an interaction between unstructured and viral mRNA with eIF3g and eIF4A. This crosslinking occurs in the same mRNA region that crosslinks to the ES6S region suggesting that eIF3g and eIF4A are located in this area in the 48S PIC. However, only eIF3g is found to crosslink to this region on the unstructured mRNA, raising a question about how general the position of eIF4A may be on non-viral mRNAs.

We previously found that eIF4A crosslinking with mRNA was only detected on mRNAs bearing a stable SL within the 25-35 nt stretch downstream the AUGi, suggesting that the presence of stable SL jams eIF4A´s helicase activity, allowing the snapshot of eIF4A-mRNA interaction. No eIF4A crosslinking was detected when SL of mRNA was removed, or when experiments were programmed with unstructured (flat) mRNA, suggesting that the presence of stable structures downstream the AUG, rather the viral or non-viral origin of mRNAs, allowed the detection of eIF4A-RNA interaction. Thus, a pattern of protein crosslinking similar to that found with SV and SFV DLP mRNAs was detected when an mRNA based on β-globin was used (Toribio et al., 2018, Figure S1).

The second major finding is that perturbation of the ES6S region, either by an oligo-fluorophore conjugate that is complementary to ES6Sd or by an EGFP fused eS4 protein (located near ES6S), leads to reduced translation rates. This is shown using reporter mRNAs as well as on a genome wide scale. Interestingly, inhibition of translation is evident on mRNAs harboring highly structured and/or long 5'UTRs (including G quadruplex containing 5'UTRs). The specificity of inhibition is controlled for by using the same oligo without the fluorophore or to an oligo that is complementary to a different rRNA region.The main weakness of the study is that the authors have not directly shown whether the oligo conjugate and EGFP-eS4 actually inhibits the interaction of the mRNA and the ES6S region. All interpretations of the experiments assume this, but this should be directly tested using the 4-thio-U crosslinking experiment in the presence of the oligo and EGFP-eS4.

To confirm that VIC-oligo 4 perturbs the interaction of mRNA with ES6S region, we carried out crosslinking experiments in the presence of VIC-oligo 4.As shown now in panel (b) of Figure 3—figure supplement 4, the presence of VIC-oligo 4 drastically reduced the crosslinking of eIF3g/eIF4A bands on both SFV DLP and unstructured mRNAs. Interestingly, we found that formation of 48S complex on both mRNAs was enhanced by blocking ES6S with VIC-oligo 4, a finding that is fully consistent with data shown in Figure 7. This result not only confirms that mRNA interacts with elements of ES6S region, but also that ES6S blocking accelerates the formation of 48S complex on mRNAs with unstructured 5´UTR.

The authors monitor rate of luciferase translation in the presence of oligo and hippuristanol in Figure 7. The delay in appearance of LUC protein and the synergistic inhibition is interpreted to mean that scanning rates are slower. This seems a bit of a strong statement for an assay that only monitors the rate of luciferase translation. Without additional evidence that more rigorously tests the rate of "scanning", the authors should significantly tone down the interpretation of this experiment (that mRNA threading into the ES6S region makes scanning slower but more processive).This reviewer doesn't really see strong evidence for mRNA "threading" model from this study. The claim that eIF4A may function at the ES6S region to unwind duplex regions also seems over interpreted – as far as I can see, the eIF4A crosslink only seems to occur in the presence of viral mRNAs.

The statement that mRNA threading into the ES6S region makes scanning slower but more processive is based not only on results fromin vitro FTT experiments, but also on data using reporter mRNAs with 5´UTR of different lengths and genome-wide polysome profiling. In our opinion, the mRNA threading concept is the simplest and most plausible explanation to the results described here. The new data included in Figure 3—figure supplement 4 further supports this idea. Anyway, to be more conservative, we replaced the term threading for binding in the title and in the summary of the manuscript.

[Editors' note: further revisions were requested prior to acceptance, as described below.]

The manuscript has been improved but there are some remaining issues that need to be addressed before acceptance, as outlined below: There is one concern that was not adequately addressed. The authors state in their rebuttal that the St. John lab antibody is against eIF4AI according to the company's datasheet. […]However, the authors need to demonstrate that the antibody is specific to eIF4AI. Also, this information is not presented in the material and methods section and they keep refereeing to eIF4A throughout the paper.It is important to address this issue because the existing confusing literature that claims that eIF4AII exhibits an antagonistic function to eIF4AI muddles the literature. The authors should alert the readers to this.

To clear up doubts about the specificity of the eIF4A1 antibody used in our manuscript (STJ27247 from St. John's lab), we have silenced the expression of human eIF4A1 gene in HeLa cells. A permanent silencing using lentivirus expressing shRNAs was impossible in our hands, probably because eIF4A1 is essential for cell proliferation. So, we carried out a transient silencing by means of esiRNA specific for human eIF4A1 mRNA, and the resulting cell extracts were probed with the antibody cited above. As shown in the new panel b of Figure 2—figure supplement 1, a reduction in the intensity of the 48 kDa protein band was observed, being consistent with the observed transfection efficiency (60-70%). This result shows that STJ27247 antibody is specific for eIF4A1. We have modified the text (figure legend of figure supplement and Materials and methods section) accordingly.